# A complete telomere-to-telomere assembly of the maize genome

Jian Chen[1,8], Zijian Wang[1,8], Kaiwen Tan ●[1], Wei Huang[1], Junpeng Shi[1], Tong Li ●[1], Jiang Hu[2], Kai Wang[2], Chao Wang[2], Beibei Xin[1], Haiming Zhao[1], Weibin Song[1], Matthew B. Hufford ●[3], James C. Schnable ●[4], Weiwei Jin[1] & Jinsheng Lai ●[1,5,6,7] ✉

A complete telomere-to-telomere (T2T) finished genome has been the long pursuit of genomic research. Through generating deep coverage ultralong Oxford Nanopore Technology (ONT) and PacBio HiFi reads, we report here a complete genome assembly of maize with each chromosome entirely traversed in a single contig. The 2,178.6 Mb T2T Mo17 genome with a base accuracy of over 99.99% unveiled the structural features of all repetitive regions of the genome. There were several super-long simple-sequence-repeat arrays having consecutive thymine–adenine–guanine (TAG) tri-nucleotide repeats up to 235 kb. The assembly of the entire nucleolar organizer region of the 26.8 Mb array with 2,974 45S rDNA copies revealed the enormously complex patterns of rDNA duplications and transposon insertions. Additionally, complete assemblies of all ten centromeres enabled us to precisely dissect the repeat compositions of both CentC-rich and CentC-poor centromeres. The complete Mo17 genome represents a major step forward in understanding the complexity of the highly recalcitrant repetitive regions of higher plant genomes.

Genome sequencing has been fundamental for the advancement of many aspects of basic biology as well as medical and agricultural applications. Followed by the decoding of the very first eukaryotic genome, the 12 Mb yeast nuclear genome in 1996 (ref. 1), numerous draft genomes with varied extents of completeness have been reported, including fruit fly[2], Arabidopsis[3], human[4,5], mouse[6] and the model crop species rice[7] and maize[8]. All these pioneering genome sequencing projects were based on classical Sanger sequencing technology. Due to cost and read-length limitations, these earlier reported draft genomes typically had tens of thousands of gaps in the pseudochromosomes. For instance, the first reported maize B73 inbred line genome had more than one hundred thousand gaps, with each tiling bacterial artificial

chromosome sequence on average having more than ten gaps[8]. The development of long-read, single-molecule DNA sequencing technologies substantially improved genome assembly quality[9–11]. Using PacBio sequencing, both the improved B73 genome[12] and the Mo17 genome[13] achieved a contig N50 of more than 1 Mb, leaving only a few thousand gaps for this highly complex crop genome. Benefitting from sequencing technology advances, the entire human X chromosome and chromosome 8 (chr8) have been completely assembled from telomere to telomere without gaps[14,15], and a completely assembled human genome has recently been released[16]. The assembly of the entire chr3 and chr9 of the maize line B73-Ab10 has also been reported[17]. For the relatively compact Arabidopsis[18] and rice[19–21] genomes, high-quality

[1]State Key Laboratory of Maize Bio-breeding, National Maize Improvement Center, Frontiers Science Center for Molecular Design Breeding, Department of Plant Genetics and Breeding, China Agricultural University, Beijing, P. R. China. [2]Grandomics Biosciences, Wuhan, P. R. China. [3]Department of Ecology, Evolution, and Organismal Biology, Iowa State University, Ames, IA, USA. [4]Department of Agronomy and Horticulture, University of Nebraska–Lincoln, Lincoln, NE, USA. [5]Center for Crop Functional Genomics and Molecular Breeding, China Agricultural University, Beijing, P. R. China. [6]Sanya Institute of China Agricultural University, Sanya, P. R. China. [7]Hainan Yazhou Bay Seed Laboratory, Sanya, P. R. China. [8]These authors contributed equally: Jian Chen, Zijian Wang. ✉e-mail: jlai@cau.edu.cn

gap-free reference genomes have also been achieved, with only several chromosomal ends, including nucleolus organizer regions (NORs), remaining incomplete. Similarly, an essentially completed banana genome of 485 Mb has been reported[22], while the watermelon genome, with a size of 369 Mb and only several dozens of 45S rDNA copies, has been completely assembled[23].

Maize is an important crop worldwide. Having a genome size very close to that of humans, while containing over 80% repetitive sequences, maize is also known as a model for complex genomes. The inbred lines B73 and Mo17 are the parental lines of one of the best-performing early commercial hybrids and of the most widely used bi-parental genetic mapping population[24,25]. Therefore, the generation of their genome sequences is of great significance[8,12,13]. Due to the rich history of genetic studies in maize and its exceptional intraspecific genome diversity[26–29], several additional inbred lines have also been sequenced, including the key Iodent line PH207 (ref. 30), the tropical lines SK[31] and K0326Y[32], European inbred lines F7, EP1, DK105 (ref. 33), the sweet corn Ia453 (ref. 34), the genetic transformation competent line A188 (ref. 35) and the inbreds underlying classical genetic studies W22 (ref. 36) and B73-Ab10 (ref. 17). More recently, high-quality genome sequences of 25 nested association mapping (NAM) founder lines and a further improved B73 assembly (version 5) have also been reported[37]. Yet, with the exception of B73-Ab10 with 53 gaps, all other assembled maize genomes reported still have hundreds or thousands of unfilled gaps. Here we report the complete telomere-to-telomere (T2T) Mo17 genome using a combination of ultralong Oxford Nanopore Technology (ONT) and PacBio HiFi reads, which marks a major step forward for genome assembly and uncovers the recalcitrant structural features of the highly complex maize genome.

## Results

### T2T assembly of all ten chromosomes of the maize Mo17 genome

We generated a total of 237.7× sequence coverage of raw ultralong ONT data and 69.4× coverage of PacBio HiFi data for assembling the maize Mo17 genome (Fig. 1a and Supplementary Table 1). Only the longest 332.1 Gb (153× coverage, read N50 of 100.7 kb; Supplementary Table 1) of the high-quality, ultralong ONT reads were selected for genome assembly. The initial ONT assembly, a total of 2.42 Gb and containing 567 contigs (Supplementary Table 2), was iteratively polished by ONT and PacBio HiFi data generated here, as well as Illumina PCR-free data generated previously[13] (Supplementary Table 1). With a high-density genetic map[38], 19 contigs that were nonredundant with each other and could cover all remaining 548 contigs were anchored and oriented onto ten pseudomolecules (Extended Data Fig. 1), including one contig that was split into two due to assembly error (Supplementary Fig. 1). Overall, using ultralong ONT reads, we obtained the basal Mo17 assembly, including ten pseudomolecules with only ten gaps, with each of chr3, chr4, chr5, chr7 and chr10 covered by one single contig (Fig. 1b).

We then assembled the Mo17 genome based on 69.4× PacBio HiFi data and integrated it into the basal Mo17 assembly to close gaps and correct assembly errors of the ONT-based assembly (Supplementary Text, Extended Data Figs. 2–4 and Supplementary Figs. 2–9). In total, six genomic regions (all were low-coverage regions (LCRs)) with assembly errors were identified by analyzing ONT reads-based alignments. Using the PacBio contigs, five gaps were closed and five misassembled LCRs were corrected for the ONT-based assembly (Fig. 1c,d and Extended

Data Figs. 2 and 3), including the 556.4 kb rDNA array on chr2, which harbored 1,387 5S rDNAs (Fig. 2a). The only uncorrected LCR (LCR6) was split forming a new gap (termed as gap_LCR6; Supplementary Fig. 3). The PacBio contigs were also used to finish the assembly of the end of the short arm of chr2 (chr2S), the only incomplete telomeric region in the ONT-based assembly (Supplementary Text and Supplementary Fig. 9). To avoid telomeric repeat sequences being incorrectly trimmed by the assembler, we further corrected telomeric regions using ONT reads and, thus, obtained the assemblies of the ends of all ten chromosomes (Fig. 1e and Extended Data Fig. 4).

Following gap-closing and correction by the PacBio assembly, there were only six gaps remained, including five related to super-long thymine–adenine–guanine (TAG) repeat arrays on chr1 (gap1 and gap2), chr2 (gap3 and gap5) and chr4 (gap_LCR6) and one related to the 45S rDNA array on chr6 (gap6; Supplementary Text). For the five super-long TAG repeat array-related gaps, two of which were closed by manual extension using the ultralong ONT reads, including 375 kb gap1 which 67.64% of sequences were TAG repeats and 1.56 Mb Gap5 which included 890.9 kb TAG repeats (Fig. 1f and Extended Data Fig. 5). Gap2, gap3 and gap_LCR6 were also extended around 700 kb, 100 kb and 1 Mb, respectively. However, each of these still included one region (with only TAG repeats, allowing few point mutations) that was not spanned by ONT reads, in which the length of two (corresponding to gap3 and gap_LCR6) were determined by the BioNano molecules[13] and one (corresponding to gap2) was determined using ONT data (Extended Data Fig. 5 and Supplementary Fig. 10). Thus, gap2, gap3 and gap_LCR6 were closed, with lengths of 637.39 kb, 211.26 kb and 1.13 Mb, respectively (Supplementary Text and Extended Data Fig. 5). With the exception of gap3, in which TAG repeat array had no transposable element (TE) insertion, about 33% sequences of the TAG arrays for gap1, gap2, gap5 and gap_LCR6 were TEs, consisting mainly of Gypsy elements (Supplementary Table 3). The assembly accuracy of these super-long TAG repeat arrays was also validated using the fluorescence in situ hybridization (FISH) assay, in which the intensities of the six most remarkable FISH signals were consistent with that of our assembly (Fig. 2b). The 45S rDNA related gap, with a total length of 26.8 Mb containing 2,974 45S rDNAs, was closed by PacBio HiFi reads based on sequences diversity among different rDNA copies (Supplementary Text, Fig. 1g and Extended Data Fig. 6). The copy number of 45S rDNA assembled was highly consistent with that estimated by four independent approaches (Fig. 2c). Finally, we obtained a complete T2T assembly of all ten chromosomes of the maize Mo17 genome, with a total size of 2,178.6 Mb (Table 1).

### Extensive evaluation of the final T2T Mo17 assembly

The overall base accuracy of the T2T Mo17 assembly was estimated to be 99.994% (quality value score, 42.33) based on mapped k-mers using Illumina PCR-free data[39]. We remapped the ultralong ONT and PacBio HiFi reads to the T2T Mo17 assembly and found uniform coverage across nearly all genomic regions, which confirmed the overall accuracy of the assembly (Supplementary Text, Extended Data Fig. 7 and Supplementary Fig. 11). Based on analysis of 7,751,268 quality-passed ONT reads longer than 10 kb, we did not detect any reads that originated from the Mo17 genome but failed to map to the final assembly (Supplementary Text and Extended Data Fig. 8), confirming the completeness of our assembly. The published Merqury's k-mer completeness metrics[39] was also applied. Approximately 99.92% of k-mers identified by Illumina

---

**Fig. 1 | Telomere-to-telomere assembly of the Mo17 genome. a**, Plant and ear photos of Mo17. **b**, Whole-genome coverage of ONT reads across the basal Mo17 assembly. Ultralong ONT reads longer than 10 kb were used for coverage analysis. The LCRs with reads depth lower than 100 and high-coverage regions (HCRs) with reads depth higher than 250 were marked by black shades. **c,d**, Schematic representation showing that a gap (**c**) and LCR (**d**) on the basal Mo17 assembly were closed or corrected by the contigs of PacBio Hifiasm assembly, in which the validity was confirmed by uniform ONT reads coverage and tiling ONT reads.

**e**, Validation of the final assembly of the terminal 1 Mb regions for chromosome 4. Red pentagrams indicated the ONT reads used to correct the telomere length for corresponding chromosomal ends, in which the telomeric repeats harbored by them were longer than other reads mapped to corresponding ends. **f**, Schematic representation showing the manual closing for a TAG repeat array-related gap on chromosome 2 by ONT reads. **g**, Validation of the assembly of the TE-rich region in the 45S rDNA array by ONT reads. The red arrows represent the transcriptional directions of 45S rDNAs.

PCR-free and PacBio HiFi data were harbored in our assembly. For the remaining 0.08% of undetected k-mers, about 60% were introduced by base errors within reads or assembly, 30% were introduced by the accumulation of multiple similar base errors of reads from several different genomic regions (Supplementary Fig. 12) and a small amount was possible exogenous DNA contamination failed to be excluded.

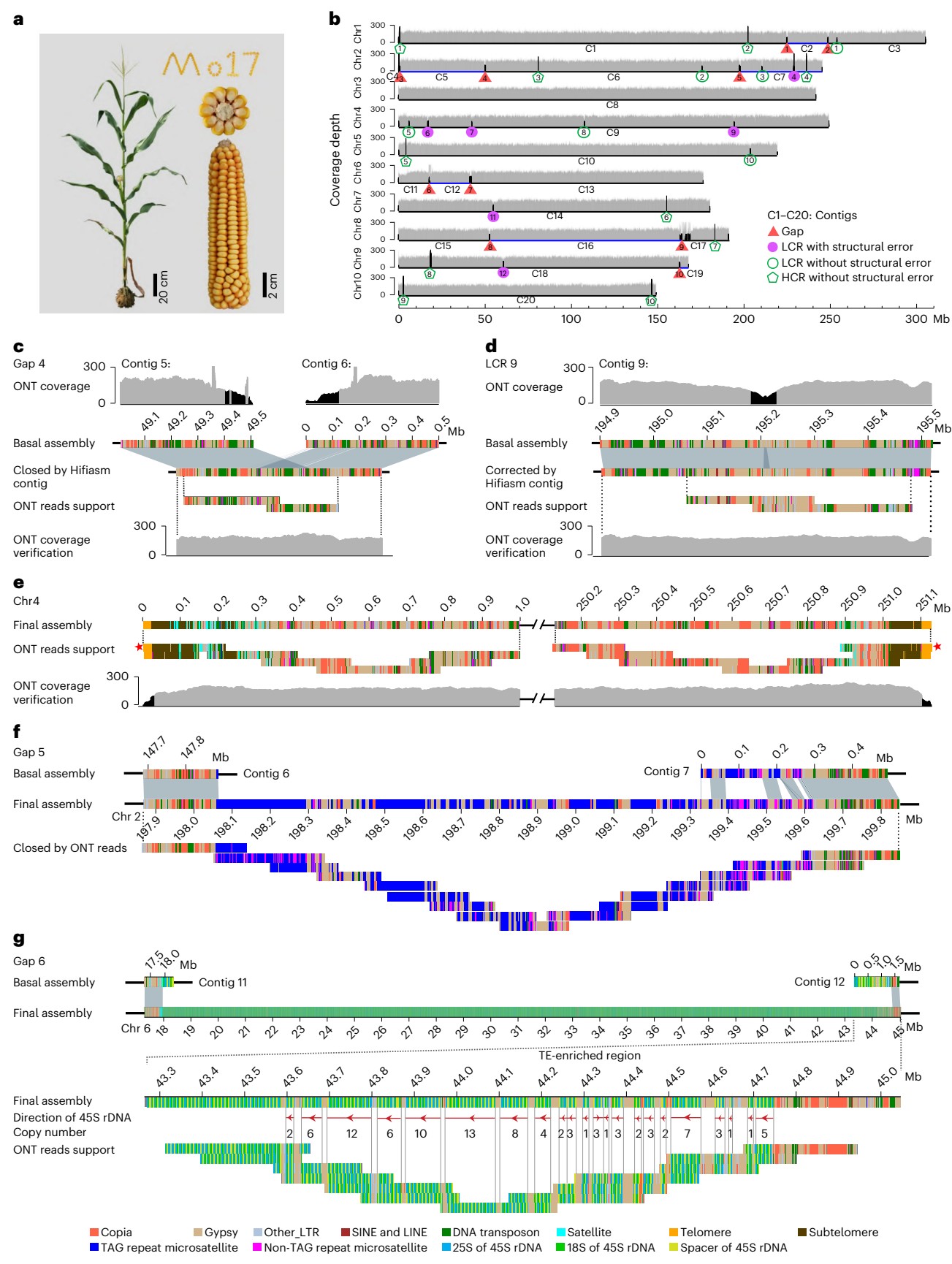

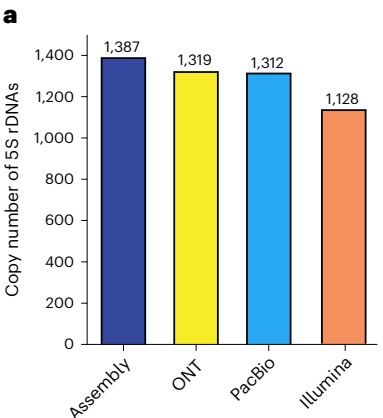

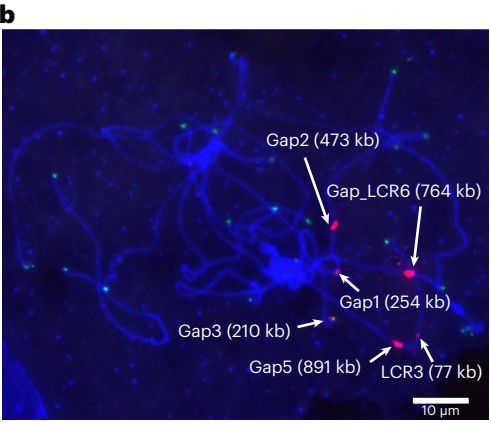

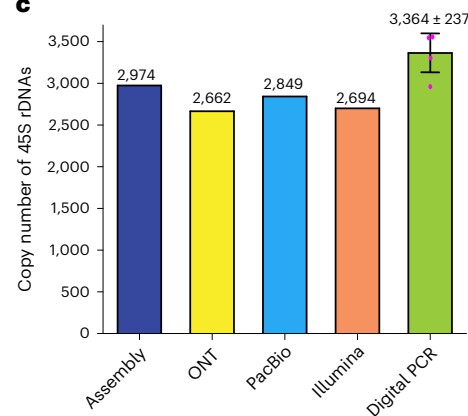

**Fig. 2 | Validation of the rDNA arrays and TAG repeat arrays. a**, Comparison of the copy number of 5S rDNAs in the assembly and that estimated with ultralong ONT, PacBio HiFi data and Illumina PCR-free data. **b**, The hybridized locations of the probes TAG repeats (red) and telomeric repeats (green) on the meiotic pachytene chromosomes of Mo17. The lengths of TAG repeats harbored by corresponding TAG repeat arrays were pointed out. Four replicates were conducted. **c**, Comparison of the copy number of 45S rDNAs in the assembly and that estimated by ultralong ONT data, PacBio HiFi data, Illumina PCR-free data and digital PCR. The data of digital PCR-based estimation were obtained from four replicates. The mean ± s.d. was represented.

## Table 1 | Summary of Mo17 assembly

| Chromosomes | Length (bp) | Gene number | New anchored genes[a] | New assembled genes[b] |
|---|---|---|---|---|
| 1 | 307,335,809 | 6,367 | 129 | 45 |
| 2 | 247,431,027 | 5,129 | 150 | 43 |
| 3 | 242,843,974 | 4,571 | 83 | 26 |
| 4 | 251,128,394 | 4,586 | 88 | 27 |
| 5 | 220,303,002 | 4,840 | 114 | 19 |
| 6 | 201,729,004 | 3,543 | 99 | 20 |
| 7 | 181,266,306 | 3,453 | 86 | 15 |
| 8 | 208,583,295 | 3,756 | 121 | 18 |
| 9 | 168,156,889 | 3,271 | 102 | 19 |
| 10 | 149,826,620 | 3,064 | 57 | 14 |
| Total | 2,178,604,320 | 42,580 | 1,029 | 246 |

[a]Genes that were not included in the pseudomolecules of the Mo17ref_V1 (ref. 13). [b]Genes that were not included in the Mo17ref_V1 assembly[13].

### Genome annotation

The T2T Mo17 includes gapless T2T assemblies for all ten chromosomes, which is an important advance as compared with previous incomplete assemblies of maize genomes, including recently reported genomes of B73 and 25 NAM founder lines with hundreds of gaps and several unfinished chromosomal ends[37]. About 88.37% of the Mo17 genome was annotated as repetitive elements, including 75.52% retrotransposons and 9.78% DNA transposons (Supplementary Table 4). Using a sliding window method, 13 TE arrays (>95% sequences were TEs with no additional genes) longer than 700 kb were identified, with the longest (1.18 Mb) array found from 79.39 to 80.57 Mb on chr3 (Supplementary Fig. 13). In addition, we identified 5.45 Mb of microsatellites, 16.43 Mb of minisatellites, 36.98 Mb of satellites, 0.44 Mb of 5S rDNAs and 26.08 Mb of 45S rDNAs, which collectively accounted for 3.92% of the genome (Supplementary Text and Supplementary Table 4). Totally, this complete assembly added (~85%) or corrected (~15%) 127.15 Mb of the Mo17ref_V1 (ref. 13; Supplementary Table 4).

Gene annotation of the Mo17 genome was performed using GeMoMa[40], Mikado[41], PASA[42] and MAKER[43] with evidence from protein homologies, RNA-seq and/or ISO-seq, complemented with ab initio Fgenesh[44] prediction. Gene models predicted by different approaches were combined by EvidenceModeler[45] to obtain an optimal nonredundant set of gene annotations (Supplementary Fig. 14). After removing transposon genes, we obtained a total of 42,580 high-confidence, protein-coding genes (Supplementary Text and Supplementary Table 5), of which 1,029 were newly anchored, including 246 newly assembled genes (Table 1 and Supplementary Fig. 15). Gene amplifications, including tandem gene duplications, have an important role in genome evolution. Among genes annotated in the Mo17 genome, 1,209 loci (including 2,916 genes) were found to have amplified through tandem duplication. Most of these loci (1,165) were moderately amplified with local copy number less than 5. One extreme case was an approximate 800 kb newly assembled region between *Zm00014ba065330* and *Zm00014ba065630* on chr10, which contained a total of 29 genes and six putative pseudogenes (Supplementary Text and Extended Data Fig. 9).

Segmental duplication is another important origin of gene duplication in maize. Using a length threshold of >1,000 bp and alignment identity higher than 98%, 55,189 nearly identical segmental duplications were identified, with an average size of 6,236 bp and a total size of 344.16 Mb. These nearly identical segmental duplications contained 1,679 nearly identical paralogous gene groups relating to 2,062 annotated genes and 4,693 unannotated putative pseudogenes, which accounted for about 5% of all annotated genes, fivefold higher than the proportion of nearly identical paralogs estimated previously[46]. Each paralogous gene group contained an average of four sequence copies (including both annotated genes and unannotated putative pseudogenes). Most (~85%) of near-identical paralogous copies were dispersed across chromosomes, with only about 15% of copies appearing as tandem duplicates. For segmental duplications without genes, the proportion of tandem segmental duplication is about 12%.

### Genome structure of most abundant satellite arrays

Knob180 (total 23.47 Mb), CentC (total 7.07 Mb) and TR-1 (total 2.86 Mb) repeats were the three most abundant satellites and comprise 90.32% of the length of satellites identified. By contrast, only 5.60% of knob180, 14.74% of TR-1 and 4.13% of CentC repeats were assembled in the Mo17ref_V1 (ref. 13). The majority of these (97.62%) were included in 25 knob180, 17 TR-1 and 17 CentC arrays (Supplementary Tables 6–8). In contrast to CentC arrays that were all located in centromeric and pericentromeric regions, knob180 and TR-1 arrays were mainly found in the chromosome arms of chr1L, chr4L, chr6S, chr6L, chr8L and chr9S. Consistent with earlier cytological work[47], 14 of 17 TR-1 arrays overlapped with knob180 arrays.

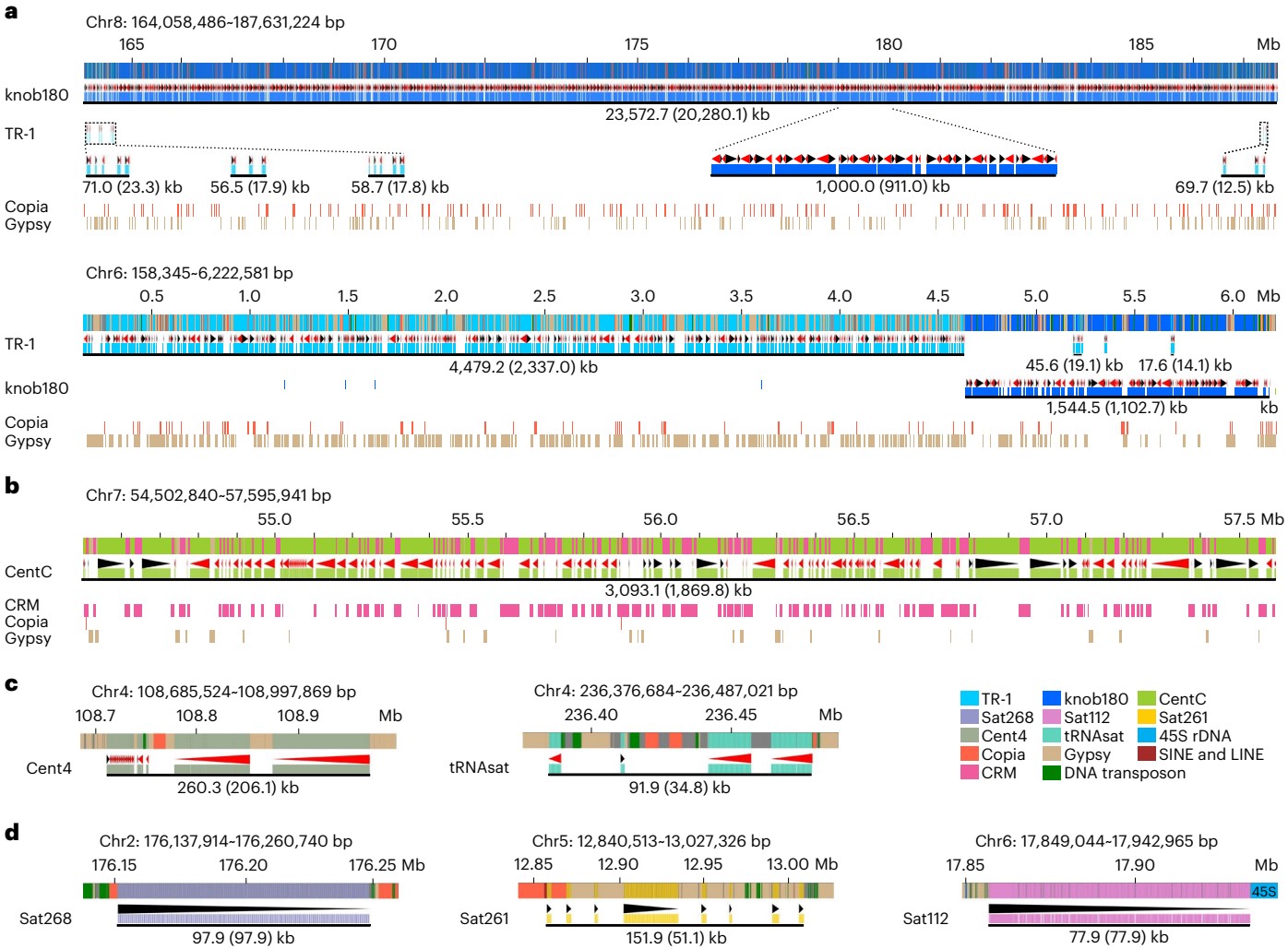

**Fig. 3 | Genome structure of satellite arrays. a**, Genome structure of two knobs on chromosomes 6 and 8. **b**, Genome structure of the longest CentC array. **c**, Genome structure of Cent4 and tRNAsat arrays. **d**, Genome structure of Sat266, Sat261 and Sat112 arrays. The length of the corresponding array was indicated below the black solid lines, in which the number in the brackets indicates the length of corresponding satellites harbored in the array. The black and red triangles under the satellites indicate the sequence direction of the corresponding satellites.

The two most substantial knobs were found on chr8L (knob-8L) and chr6S (knob-6L) and together accounted for 90.86% and 92.52% of TR-1 and knob180 repeat copies, respectively. Knob-8L, 23.57 Mb in total, had 115,117 knob180 repeats, which accounted for 86.03% of sequences of knob-8L. TR-1 repeats, 235 copies in total, were contained in four TR-1 arrays less than 25 kb in the terminal regions of knob-8L and made up 0.3% of knob-8L (Fig. 3a). The remaining 13.2% of knob-8L consisted of TEs, including Gypsy (47.35%) and Copia (44.85%) elements. Knob-6S, 6.03 Mb in total, was composed of a 4.48 Mb TR-1 array (including 10,564 TR-1 copies) along with a 1.54 Mb knob180 array (including 6,107 knob180 copies; Fig. 3a). About 45.85% of sequences of the TR-1 array and 25.22% of the sequences of the knob180 array on knob-6S were TEs, which were 85.74% Gypsy elements. TEs were spread evenly in knob-8L, knob-6S (Fig. 3a) and CentC arrays (Fig. 3b), with no particular TE-poor or TE-rich regions along the satellite arrays. Interestingly, there were two types of knob180 repeats according to their variant distances (Supplementary Text). Knob180 repeats with relatively higher variant distances were substantially enriched on Knob-8L (Extended Data Fig. 10).

In addition to knob180, TR-1 and CentC arrays, there were five notable satellite arrays in the Mo17 genome (Supplementary Table 9), including a 260.29 kb Cent4 array in the pericentromeric region of chr4 and a 91.95 kb tRNA satellite array on chr4L, which had both been identified previously[48,49] (Fig. 3c). Sat268 (repeat unit, 268 bp) array on chr2L, sat261 (repeat unit, 261 bp) array on chr5L and sat112 (repeat unit, 112 bp) array on chr6S were all newly discovered, with length of 176.33 kb, 151.88 kb and 77.86 kb, respectively (Fig. 3d). No homologous sequence of the sat268 repeat was identified in B73 and 10 of 25 NAM founder lines[37]. This presence/absence variation might have been generated in their ancestors as only three of 19 wild relatives were found with sat268 repeats according to resequencing data from maize Hapmap2 (ref. 50).

## Genome structure of completely assembled rDNA arrays

The number of rDNA loci can be variable for different species[51]. In maize, the 5S rRNA resulted from the transcription of 5S rDNAs and the 18S, 5.8S and 25S rRNAs are produced by splicing of a single 45S transcript encoded by 45S rDNAs (Fig. 4a). Consistent with a previous fluorescence in situ hybridization (FISH) study[52], only one 5S rDNA array and one 45S rDNA array were detected on chr2L and chr6S, respectively (Supplementary Table 10), and no additional intact 5S and 45S rDNA copies were detected in other genomic regions.

Based on 48 SNPs and 22 indels identified among different 5S rDNA copies (Fig. 4b), the 5S rDNAs could be divided into 346 genotypes.

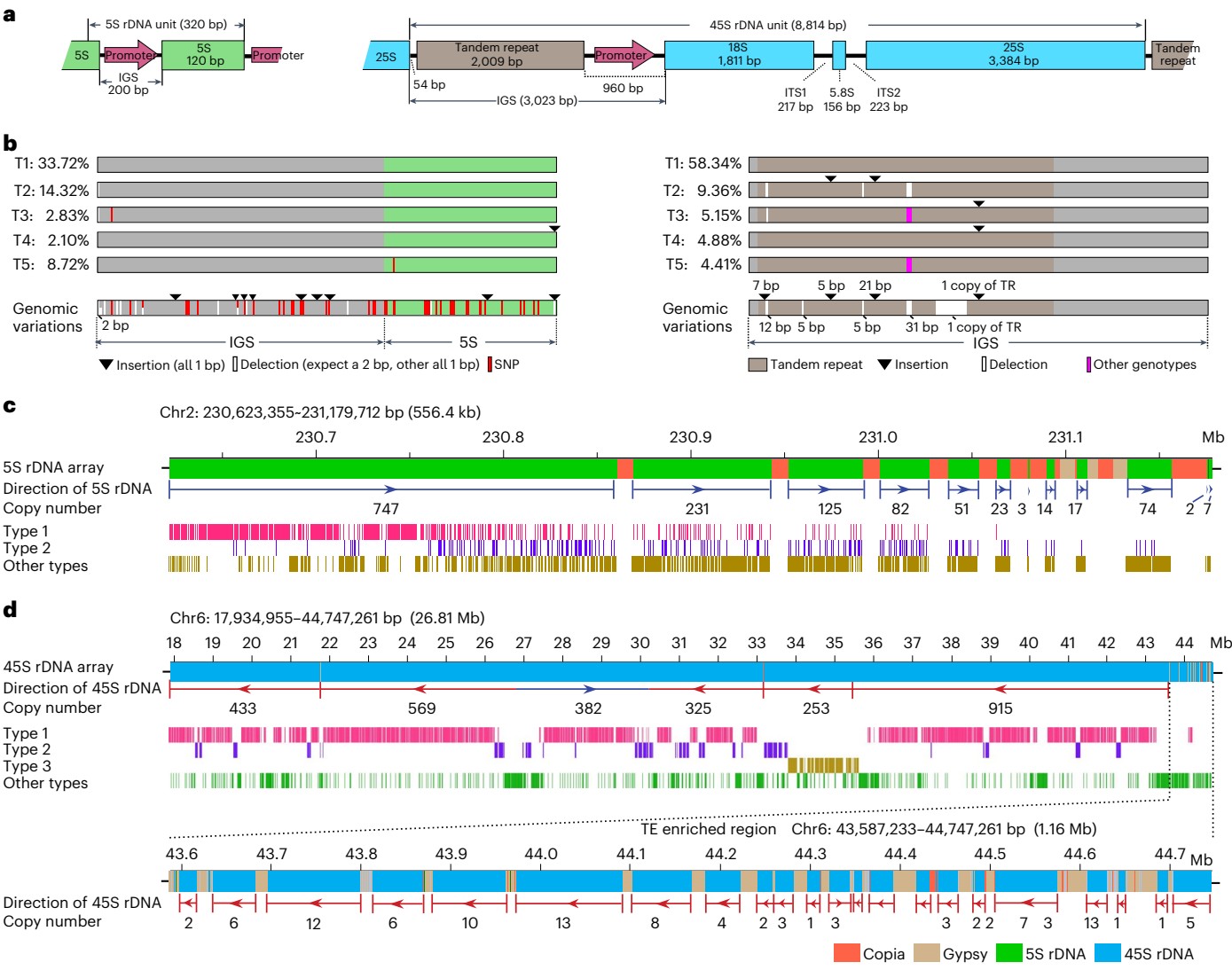

**Fig. 4 | Genome structure of 5S and 45S rDNA arrays. a**, Sequence structure of a typical 5S and 45S rDNAs repeat unit. IGS, intergenic spacer region; ITS, internal transcribed spacer region. **b**, Graphical representation of the five most abundant genotypes of 5S rDNAs (left) and 45S rDNAs (right). For the 45S rDNA, only the IGS region was shown. Genomic variations used for genotype analysis were indicated. **c,d**, Graphical representation of the genome structure of 5S rDNA array (**c**) and 45S rDNA arrays (**d**).

The two most common genotypes (termed type 1 and type 2) comprised 33.72% and 14.32% of 5S rDNA copies, respectively (Fig. 4b). The 5S rDNAs on the centromere-proximal end of the array might be newly generated as most were type 1, and there was no TE insertion among them (Fig. 4c). By contrast, the 5S rDNAs on the centromere-distal end might have been generated earlier as they were more diverse, with more TE insertions found in the region (Fig. 4c). All 5S rDNA copies were amplified and arranged in the same orientation, with transcription direction toward the end of chr2L (Fig. 4c).

There were 64 genotypes for 45S rDNAs based on nine indels (longer than 5 bp) identified within the intergenic spacer, with the most common genotype accounting for 58.34% of copies (Fig. 4b). Most (87.16%) of 45S rDNAs were located toward the end of chr6S as judged by their transcriptional direction, except for one 3.42 Mb inversion stretch in the middle of the array (Fig. 4d). There were 54 interspersed sequences longer than 1 kb in this 26.81 Mb NOR region, most (50) of which were TE insertions located at the 1.16 Mb centromere-proximal end. Only four TE insertions were found in the rest of the 25.66 Mb region of the array (Fig. 4d). Distinct from that TEs in the 5S rDNA array were mostly Copia, TEs in 45S rDNA array

were enriched with Gypsy elements (Supplementary Text, Fig. 4c,d and Supplementary Fig. 16), reflecting the potential impact of local genomic structure on TE insertion. Overall, our results demonstrated highly complex patterns of rDNA duplications and transposon insertions across the NOR.

## Genome structure of all ten centromeres
A typical feature of centromeres is that their DNA is bound by special nucleosomes with centromeric histone H3 (CENH3)[53]. Here we determined the centromeric regions of the Mo17 genome according to the enrichment level of CENH3 obtained via chromatin immunoprecipitation sequencing with the anti-CENH3 antibody. The average length of ten centromeres was 2.22 Mb, with the longest on chr2 (2.93 Mb) and the shortest on chr1 (1.62 Mb; Supplementary Table 11 and Fig. 5a). Overall, there was no association between centromeric length and chromosomal size, in contrast to previous observation on several chromosomes from some maize NAM founder lines[54]. The relative positions of centromeres varied among different chromosomes, with the minimum and maximum L/S ratio (long arm length/short arm length) being 1.11 (chr5) and 3.08 (chr8), respectively (Fig. 5b).

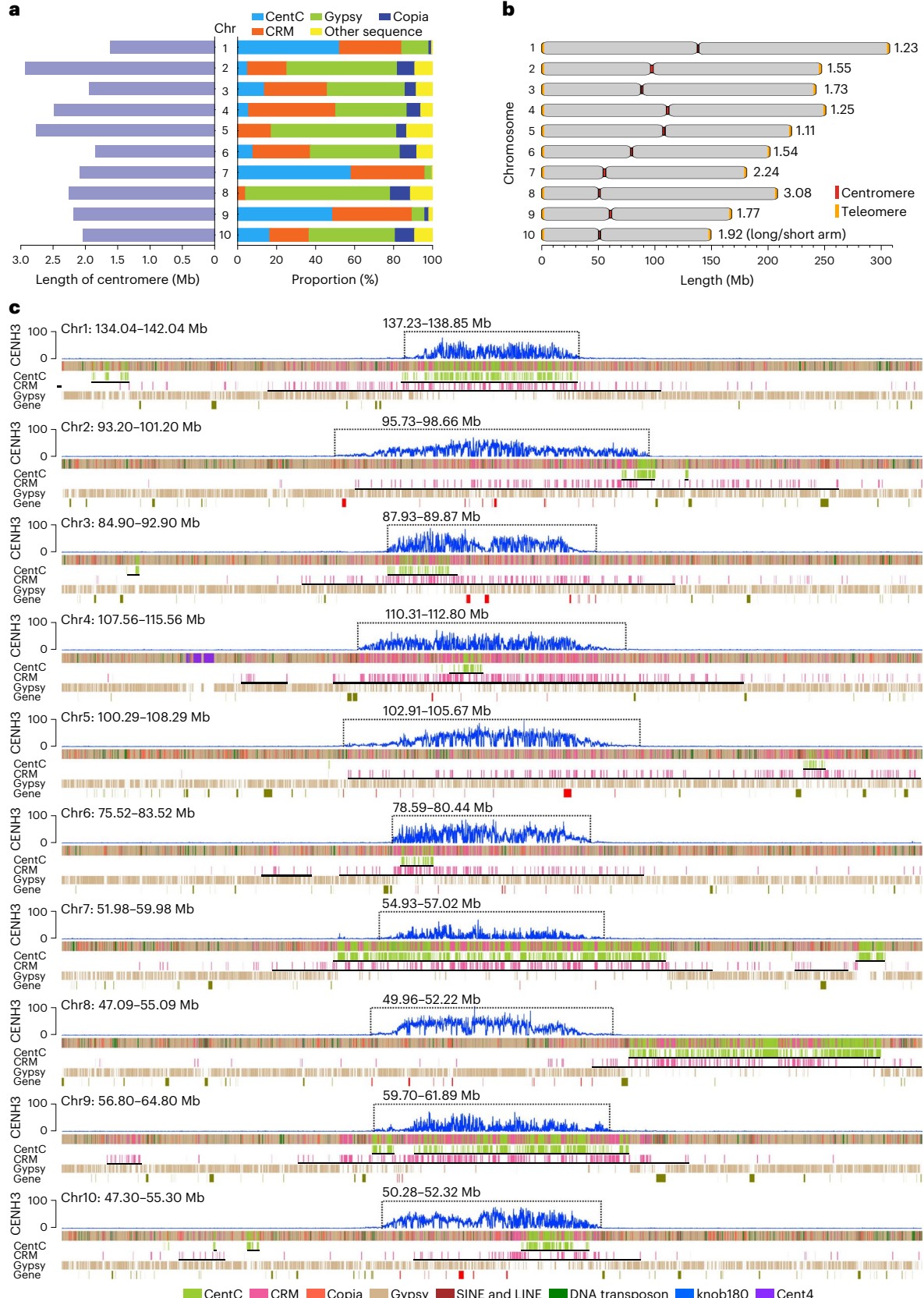

**Fig. 5 | Genome structure of the centromeric regions of ten chromosomes.**
**a**, Comparison of the length and sequence composition of ten centromeres.
**b**, Graphical representation of the centromeric positions on ten chromosomes.
**c**, Schematic representation showing the distribution of different sequence
compositions across ten centromeres. The CENH3 levels were represented by
the enrichment level in 10 kb windows along chromosomes. The centromeres
were marked by dotted boxes. The black solid lines under the tracks of CentC
and CRM indicated the corresponding regions were identified as CentC arrays or
CRM arrays. The red blocks for the track of gene indicated genes located in the
centromeres.

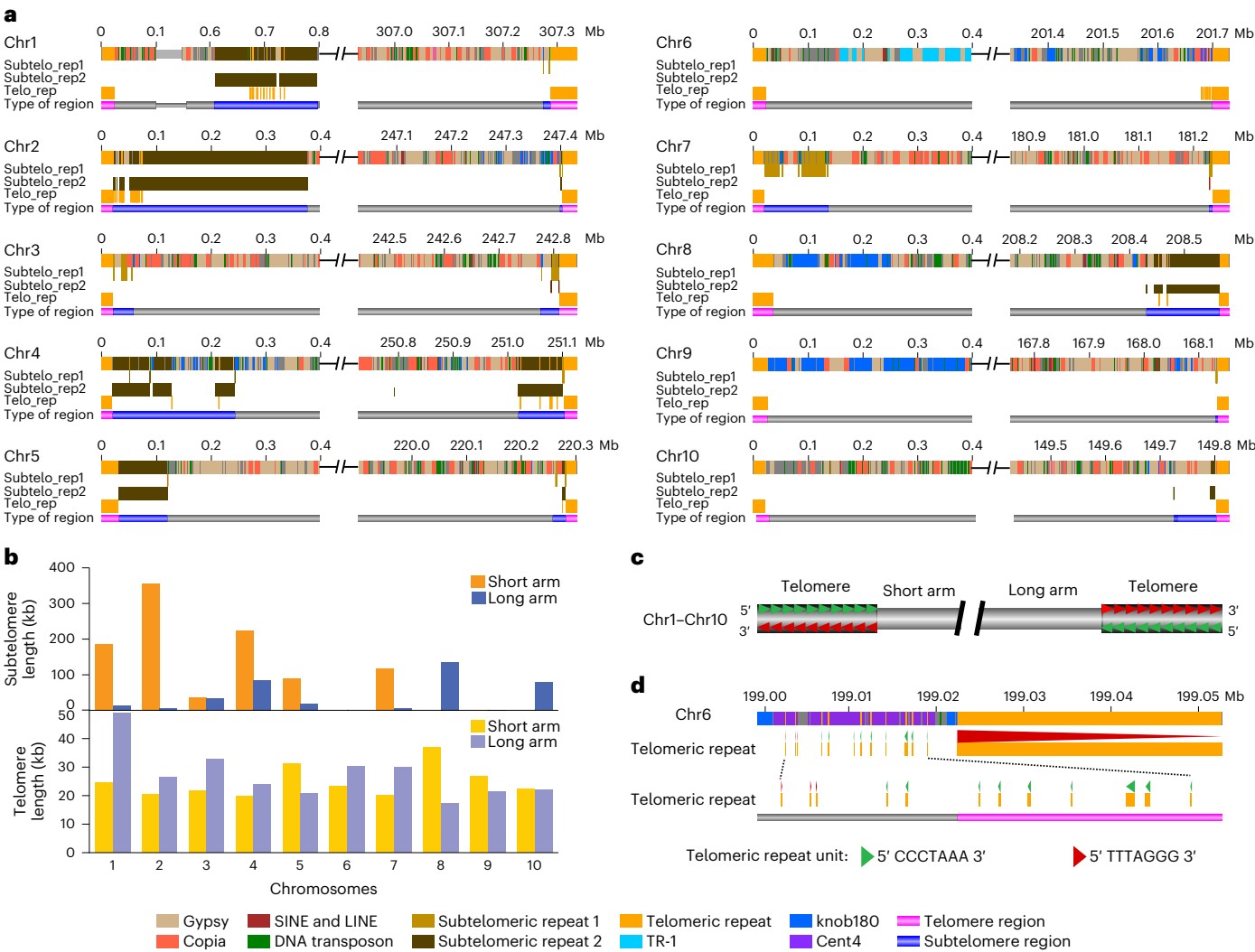

**Fig. 6 | Genome structure of the telomeric and subtelomeric regions of 20 chromosomal ends. a**, Schematic representation showing the distribution of different sequence compositions across the terminal regions of ten chromosomes. **b**, Comparison of the length of telomeres and subtelomeres for ten chromosomes. **c**, Graphical representation of the direction of telomeric repeats observed for all telomeres. **d**, The direction of telomeric repeats on the ends of chr6L.

An average of 87.3% of sequences of centromeres consisted of CentC, CRM and other non-CRM Gypsy retrotransposons, including Cinful-Zeon, Flip, Prem1 families of Gypsy elements (Fig. 5a and Supplementary Table 11). However, the proportions of CentC, CRM and non-CRM Gypsy in different centromeres varied greatly, ranging from 0% to 58.3%, 4.2% to 44.7% and 3.5% to 74.3%, respectively (Fig. 5a and Supplementary Table 11). The centromeres can be roughly divided into two structural types based on CentC abundance. For the CentC-rich centromeres of chr1, chr7 and chr9, almost all (average 96.87%) of their centromeric regions were covered by CentC-enriched expanses. While for the CentC-poor centromeres of chr2, chr3, chr4, chr6 and chr10, an average of only 20.25% of their centromeric regions were identified as CentC arrays, and the centromeres of chr5 and chr8 had no satellites, including CentC at all (Fig. 5c). The CentC repeat contents of the three CentC-rich centromeres (on average, 53.15%) was remarkably higher than that of the seven CentC-poor centromeres (7.03%; Fig. 5a). Centromeres were generally located in the regions with the most abundant CentC and CRM repeats of a given chromosome, with CRMs relatively evenly distributed across entire centromeric regions, while CentC preferentially appeared as clusters (Fig. 5c). According to the report that large expanses of CentC and CRM can exist outside of centromeres[54], about 28.73% of CentCs and 56.20% of CRMs were

found in pericentromeric areas rather than centromeres. An extreme case was the centromere of chr8, which lies in the upstream of a typical CentC and CRM enriched expanse (about 2.3 Mb), a region enriched with Gypsy but with few CRMs and no CentC (Fig. 5c).

A total of 82 genes were identified in centromeres of the Mo17 genome, of which 72 had homolog genes in maize NAM founder lines and 52 had homolog genes in sorghum. About half (46) of these centromeric genes were expressed with the value of fragments per kilobase of transcript per million mapped reads (FPKM) > 1), slightly lower than the ratio expressed among all annotated genes (74.8%), and that the centromeric genes relatively preferred to be tissue specifically expressed (Supplementary Fig. 17a). Interestingly, non-CRM Gypsy abundance was generally positively correlated (r = 0.722) with the number of genes in centromeres (Supplementary Text, Fig. 5 and Supplementary Fig. 17b), reflecting a potential role of non-CRM Gypsy insertions in gene content in centromeres.

## Length and sequence composition variations of subtelomeres
Telomeric sequences are often missing or far from complete in basically all current versions of assembled plant genomes. The average length of 20 telomeres of the complete T2T Mo17 genome was 26.1 kb, with about 3,700 telomeric repeat copies for each telomere (Fig. 6a), longer

than average telomere size (11.7 kb, ranging from 1.8 kb to 40.0 kb) of 22 maize inbreds estimated by Southern blotting[55]. Telomere length varied less than threefold among different chromosomal ends, with the longest on chr1S (49.0 kb) and the shortest on chr8L (17.4 kb; Fig. 6b and Supplementary Table 12). Remarkably, the G-rich strand of telomeric repeats (5'-TTTAGGG-3') at telomeres always oriented toward the ends for all ten chromosomes (Fig. 6c). This conservation of direction did not hold outside of telomeric regions as some inverted copies of telomeric repeats were observed at flanking regions of telomeres, such as that of chr6L (Fig. 6d).

In the Mo17 genome, there were 15 chromosomal ends with typical subtelomeric repeats, including subtelomeric repeat 1 (629 bp), which was homologous with previously reported maize subtelomere sequence U39642 (ref. 56), and subtelomeric repeat 2 (532 bp), which had about 170 bp homologous to maize subtelomere sequence 4-12-1 (ref. 57). The regions enriched with these subtelomeric repeats were identified as subtelomeres. They were all immediately adjacent to telomeres, except for chr1S where the telomere and subtelomere were interrupted by a 587.87 kb region with 74% TEs and 26% nonrepetitive sequences including 24 genes (Fig. 6a and Supplementary Table 13). The average length of subtelomeres was 92.57 kb, with 200-fold variation between the longest (357.13 kb, chr2S) and the shortest (1.78 kb, chr9L) subtelomeres (Fig. 6b). Extreme variation was also observed for subtelomeric sequence composition. For example, TEs occupied 87.53% of sequences of the subtelomere of chr10L but were not found for subtelomeres of chr2L, chr4L, chr5S, chr7L and chr9L. Although the proportions of TEs were similar for subtelomeres of chr3L (40.84%) and chr5L (50.76%), the former was mostly Gypsy while the latter was Copia. Moreover, strong chromosome-specific patterns were observed for subtelomeric repeats, as the proportion of subtelomeric repeat 1 was more than 70-fold higher than that of subtelomeric repeat 2 for subtelomeres of chr1L, chr3S, chr3L, chr7S, chr7L and chr9L, but was more than 30-fold lower than that of subtelomeric repeat 2 for subtelomeres of chr1S, chr2S, chr4S, chr4L, chr5S, chr8L and chr10L (Fig. 6a, and Supplementary Table 13). We noted there were five chromosomal ends without typical subtelomeric repeats (chr6S, chr6L, chr8S, chr9S and chr10S). However, when inspecting all 400 kb terminal regions of these chromosomes, with the exception of chr10S, abundant tandem repeats were found from the subtelomeric regions of all other four chromosomal ends. There were 121 kb TR-1 repeats for chr6S, while chr6L, chr8S and chr9S were enriched with knob180 repeats, with length of 28 kb, 147 kb and 274 kb, respectively (Fig. 6a). In addition, chr6L also harbored 12 kb of Cent4 repeats. Overall, our analyses of the 20 completely assembled chromosome ends revealed extensive length and sequence composition variation in subtelomeric regions.

## Discussion

Aiming to uncover the complex genomic 'dark matter' and to decode the extraordinary repetitive regions of higher plant genomes, we have here achieved a complete T2T assembly of all ten chromosomes of the maize Mo17 inbred line by combining the advantages of both the latest ONT and PacBio sequencing technologies. Due to the highly tandem duplication of the 45S rDNA, it is challenging for assembly of NOR. There were two 45S rDNA clusters at the tips of the chr2S and chr4S of Arabidopsis, which each contained about 365 rDNA copies[58]. Rice genome also has two 45S rDNA clusters at the ends of chr9S and chr10S, which together contain about 850 rDNA copies[59]. However, these 45S rDNA clusters have not been completely assembled thus far, including the latest updated Arabidopsis[18] and rice[19,20] genomes. In plants, the NOR of watermelon genome[23], with only several dozens of 45S rDNAs, has been recently assembled. Here our T2T Mo17 genome provides a completely assembled genomic structure of a megabases-scale plant NOR, which contains about 3,000 rDNA copies. According to previous studies, only a small fraction of rDNA units is transcribed[60,61]. However, no remarkable local sequence composition variation was observed

for different parts of the NOR. Transcriptional inactivation might be mainly due to the fact that the rDNAs are packaged inside the nucleolus and, thus, are less accessible for the transcription machinery when compared with the rDNAs at the nucleolar boundaries[61].

Telomeres are highly repetitive sequences found at the ends of eukaryotic chromosomes. Average assembled length of the Mo17 telomeres was 26.1 kb (16–48 kb) and are, therefore, near the longest assembled telomeres so far. The range of length variation between different telomeres is similar to the estimated telomere length variation of rice (5.1–10.8 kb)[62] and human (5–20 kb)[63]. So far, there is no rigid definition of a subtelomere, as subtelomeric sequences are fast evolving and many subtelomeric repeats are species-specific, and often chromosome-specific[64,65]. Consistent with this, we found extreme sequence variation in the 20 subtelomeres of maize, with no typical subtelomeric repeat found for five chromosomal ends. Hence, we speculate that the 'functional subtelomere' might be not only determined by sequences but also by other elements like epigenetic modifications.

Our assembly uncovered several extremely long microsatellite blocks. Much like TEs, the satellite DNA has initially been considered as junk DNA but could have actually had important structural and regulatory roles[66]. Recently, knob180 abundance and telomere length have been shown to associate with flowering time in maize[67,68]. The proportion of satellite DNA drastically varies for different eukaryotic species. In an extreme case, nearly 50% of the *Drosophila virilis* genome is satellite DNA[69]. The complete T2T Mo17 genome uncovers nearly 5% of additional repetitive sequences, many of which are the highly tandemly duplicated satellites or microsatellites. This indicates that amplification of non-TE repeats can be another important mechanism of plant genome expansion, in addition to the widely documented nested retro-transpositions[70]. To be noted that, at present, the results of most genome assembly using mixed tissues of multiple individuals reflect the sequence compositions of the majority or average of large cell populations used. Given that the highly tandemly duplicated sequences have considerable levels of copy number variation among individuals within a species[71] and even somatic instability among different tissues of each individual[51,72], it is still technically challenging to precisely dissect the exact copy number of these highly tandemly duplicated sequences at the extraordinary repetitive regions for particular tissues or cell types.

## Online content

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

## Methods

### Plant material and genome sequencing

The maize (*Zea mays*) inbred line Mo17 was grown in a greenhouse with conditions of 30 °C for 16 h under light and 25 °C for 8 h in the dark. After 14 d, the fresh young leaf tissue was collected and frozen immediately in liquid nitrogen for DNA extraction. High-molecular-weight genomic DNA prepared by the cetyltrimethylammonium bromide (CTAB) method and followed by purification with Qiagen genomic kit (Qiagen, 13343) was used for the construction of both PacBio HiFi sequencing libraries and ONT common sequencing libraries. High-molecular-weight genomic DNA prepared by the nuclei method[73] was used for the construction of ONT ultralong sequencing libraries. ONT and PacBio sequencing libraries were run on the Nanopore PromethION sequencer and PacBio SequeII platform, respectively. Detail libraries construction and sequencing methods are described in the Supplementary Methods.

### Whole-genome assembly of ultralong ONT data and PacBio HiFi data

For 428.7 Gb quality-filtered ultralong ONT data, the longest 332.1 Gb reads with an N50 of 100.7 kb were selected and used for whole-genome assembly by NextDenovo (v2.2-beta.0, https://github.com/Nextomics/NextDenovo) with the following parameters: seed_cutoff = 130k -n 355 -Q 6 -I 0.44 -S 0.18 -N 2 -r 0.37 -m 4.96 -C 93 -z 14.

For 151.1 Gb PacBio HiFi data, both Hifiasm[74] (v0.7, https://github.com/chhylp123/hifiasm) and Canu[75] (v.2.0, https://github.com/marbl/canu) were used for whole-genome assembly. The default parameters were set for Hifiasm assembly. The following parameters were set for Canu assembly: -assemble -p Mo17 -d canu_Mo17 genome size = 2.3g -pacbio-hifi CCS.fasta; grid Engine Resource Option = -pe smp THREADS -l mem_total = MEMORY; grid Options = -S /bin/bash -q all.q,ODE.q,asm.q; use Grid = true batMemory = 251 Gb.

### Whole-genome polishing

The sequences of all 567 contigs of initial ONT assembly were polished by ONT data, PacBio HiFi data and Illumina PCR-free data using Next-Polish (v1.1.0)[76] with recommended parameters. Briefly, we first used quality-filtered ultralong ONT reads and common ONT reads to polish the assembly for three rounds, setting 'task = best rewrite = yes rerun = 3' in the parameter config file. The assembly polished by ONT data was then further polished for three rounds with PacBio HiFi reads, setting 'task = best rewrite = yes rerun = 3' in the parameter config file. Next, the assembly was polished by four rounds with Illumina PCR-free reads, setting 'task = best rewrite = yes rerun = 4' in the parameter config file. Because of the high quality of PacBio HiFi reads, the contigs assembled using PacBio data were not polished. After closing all gaps, the sequences of gaps that were closed manually, along with their upstream and downstream 2 Mb regions, were extracted, and then they were iteratively polished by ONT data (three rounds), PacBio HiFi data (three rounds) and Illumina PCR-free data (four rounds) using NextPolish[76] (v1.1.0) with recommended parameters.

### Construction of pseudomolecules

The pseudomolecules were constructed based on a high-density genetic map containing maize pangenome genotype-by-sequencing (GBS) tags[38]. All 567 contigs of the initial ONT assembly were aligned to the genomic map by bwa mem[77] with default parameters. According to the alignments, 19 contigs that were nonredundant with each other and could cover all remaining 548 contigs were used to construct the pseudomolecules. Notably, one of the 19 contigs was spilt into two manually due to assembling error. The order and orientation of contigs were determined according to the physical positions of GBS tags.

### Identification of local coverage-anomalous regions on the basal Mo17 assembly

A total of 417.9 Gb quality-passed ultralong ONT reads longer than 10 kb were aligned to the basal Mo17 assembly using Minimap2 (ref. 78),

with the parameters of '-x map-ont -r 10000 -N 50'. Only primary alignments and supplementary alignments (value of FLAG in sequence alignment map (SAM) format file must be 0, 16, 2,048 or 2,064) with minimum query sequence coverage 0.85 were used for further read depth analysis. Read depth of each base was calculated by BEDTools[79] (v2.29.2) with the parameters of 'genomecov -bga -split'. Then, average depths for all 1 kb bins of the genome were calculated. The bins with depth lower than 100 and higher than 250 were defined as LCRs and high-coverage regions, respectively. The bins adjacent to each other were merged.

### Integration of the basal Mo17 assembly and PacBio assembly

The contigs of both PacBio Hifiasm and Canu assemblies were aligned onto the basal Mo17 assembly using Minimap2 (ref. 78), with the parameters of '-x asm5 -f 0.02'. According to the alignment, two types of PacBio contigs were integrated into the basal Mo17 assembly, including contigs that can span the gaps and contigs that can be used to correct the assembly errors harbored by the basal Mo17 assembly.

### Closure of the TAG repeat array and 45S rDNA array-related gaps

The 5 TAG repeat array-related gaps were manually closed based on the ultralong ONT reads, and the 45S rDNA array-related gap was closed based on PacBio HiFi reads, the methods of which are described in detail in the Supplementary Methods.

### FISH

Fresh Mo17 tassels were collected and fixed with Carnoy's solution (ethanol: glacial acetic 3:1, vol/vol). FISH assay was performed as described in ref. 80. FISH probes (TAG)$_{15}$ and (TTTAGGG)$_7$ (telomeric repeats) were labeled with digoxigenin and Cy5, respectively. The digoxigenin-labeled probe was detected by anti-digoxigenin antibody conjugated with Rhodamin (Roche). Anthers of Mo17 were squashed and staged with a phase contrast microscope. Slides with pachytene stage meiocytes were selected for the FISH assay. The Olympus BX61 epifluorescence microscope, equipped with a charge coupled device (CCD) camera (QImaging; RETGA-SRV FAST 1394), was used to capture the cytological images. The Image-Pro Plus 6.0 software (Media Cybernetics) was used to analyze the digital images.

### Copy number estimation of rDNAs

The copy number of 5S and 45S rDNAs in the genome was estimated by the BLAST-based method using both ONT ultralong and PacBio HiFi data, and the k-mer-based method using Illumina PCR-free data. In addition, the copy number of 45S rDNAs in the genome was also estimated using Naica Crystal Digital PCR System (Stilla Technologies). Detail methods are described in the Supplementary Methods.

### Validation of the T2T Mo17 assembly with ONT data

A total of 417.9 Gb quality-passed ultralong ONT reads longer than 10 kb were aligned to the T2T Mo17 assembly using Minimap2 (ref. 78), with the parameters of '-x map-ont -r 10000 -N 50'. Based on the alignment, properly mapped reads were identified with the criteria of the primary and supplementary alignment (value of FLAG in SAM format file must be 0, 16, 2,048 or 2,064) with minimum query sequence coverage of 0.85. Fused reads were identified with the criteria that the read was aligned to multiple genome regions, and the query coverage of all alignments amounted to 0.85. Symmetrical reads were identified with the criteria that the read can be roughly divided into two parts with sequences reverse complement with each other, and the two parts were aligned to the same genomic region with the query coverage together amounting to 0.85. Only properly mapped ONT reads were used for read coverage analysis. Read depth of each base of the T2T assembly was calculated with BEDTools[79] (v2.29.2) with the parameter of 'genomecov -bga -split'. Then, average depths for all 1 kb bins of the

genome were calculated. The bins with depth lower than 100 or higher than 250 were identified as local coverage anomalies. The bins adjacent to each other were merged. Except for properly mapped reads, fused reads and symmetrical reads, the remained ONT reads were further determined to be originated from organelle genomes or chimeric reads (Supplementary Methods).

## Validation of the T2T Mo17 assembly with PacBio and Illumina data

The accuracy of the final T2T Mo17 assembly was estimated from mapped k-mers via Merqury[39] (v1.1). In brief, 251 Gb Illumina PCR-free data released previously[13] were used to generate k-mer database with $K = 21$. In Merqury, every k-mer in the Mo17 assembly is evaluated for its presence in the k-mer database generated from Illumina PCR-free data, with any k-mer missing in the k-mer database as base-level 'error'. We found that of 2,178,604,120 $k$-mers, there were 2,676,840 $k$-mers detected only in the assembly, resulting in a quality value (QV) score of 42.3252, which was calculated as follows: $-10 \times \log(1-(1-2676840/2178604120)^{(1/21)}) = 42.3252$. Then, the accuracy of Mo17 assembly was derived directly from this QV score as follows: $100 - (10^{(42.3252/-10)}) \times 100 = 99.9941$. The mapped $k$-mers via Merqury[39] (v1.1) were also used to estimate the completeness of the final T2T Mo17 assembly, a detailed method of which is described in the Supplementary Methods.

PacBio HiFi reads were used to identify local coverage-anomalous regions on the basal Mo17 assembly. All 151.1 Gb PacBio HiFi reads were aligned to the T2T Mo17 assembly using Minimap2 (ref. 78) with the parameters of '-x map-pb -r 1000 -N 50'. Based on the alignment, properly mapped reads were identified with the criteria of the primary and supplementary alignment (value of FLAG in SAM format file must be 0, 16, 2,048 or 2,064) with minimum query sequence coverage 0.85, and then used for read coverage analysis. Read depth of each base of the T2T assembly was calculated with BEDTools[79] (v2.29.2) with the parameter of 'genomecov -bga -split'. Then, average depths for all 1 kb bins of the genome were calculated. The bins with depth lower than 20 or higher than 105 (genome-wide average: 65) were identified as local coverage anomalies. The bins adjacent to each other were merged.

## Repetitive elements annotation

Transposable element library (maizeTE02052020) was manually curated from the Maize TE Consortium (MTEC; https://github.com/oushujun/MTEC) and was then used to identify new TEs of the Mo17 genome that were not included in the MTEC library using the extensive de-novo TE annotator (EDTA)[81] (v1.7.0) with the following parameters: '-species maize -curatedlib maizeTE02052020'. Notably, the four types of CRMs, including CRM1, CRM2, CRM3 and CRM4, were all included in the downloaded MTEC library (maizeTE02052020). Besides this new TEs identified by EDTA, some other repetitive elements were also augmented into the MTEC library to construct the repetitive element library, including 5S rRNA monomer (Genebank ID: DQ351339.1), its1_5.8S rRNA_its2 sequences (Genebank ID: AF019817.1), subtelomere_U39642 (Genebank ID: U39642.1) and subtelomere_4-12-1 (GeneBank ID: CL569186.1), as well as SINEs, 25SrRNA, 18SrRNA, Cent4 and tRNASAT_ZM from Repbase[49]. RepeatMasker[82] (v4.1.1) was then used to discover and identify repeats in the Mo17 genome with the repetitive element library. Besides, we identified microsatellites, minisatellites and satellites in the Mo17 genome using Tandem Repeats Finder[83] (TRF, v4.09.1) with the following parameters: '2 7 7 80 10 50 500 -f -h -d -m'. For the result, redundant identification of tandem repeats and tandem repeats of less than five copies were manually removed. Resulted tandem repeats in which the lengths of repeat units were less than 10 bp, between 10 bp and 100 bp and longer than 100 bp were defined as microsatellites, minisatellites, and satellites, respectively.

Repetitive elements for given ONT reads were identified using RepeatMasker[82] similar to that used for identifying repeats in the Mo17 genome. Notably, satellites in the Mo17 genome identified using TRF[83]

were also added into the repetitive element library for the identification of repetitive elements for given ONT reads with RepeatMasker[82] (v4.1.1).

For a graphical representation of the sequence composition across a given genomic region or ONT read, different colors were assigned to different types of repeats annotated in the corresponding genomic region or ONT read. A '.bed' format file with the information of repetitive elements locations and corresponding colors was generated and then used for drawing the graphics using the perl script invoking the scalable vector graphics (SVG) module.

## Identification of TE arrays

We split all intergenic regions between genes into bins with window size of 100 kb and step size of 10 kb. Then we calculated the TE proportion of each bin using BEDTools[79]. The TE arrays were identified with a threshold that the TE proportion in the bin is over 0.95. Overlapped TE arrays were merged.

## Generation of isoform-sequencing (ISO-seq) data

To aid in genome annotation, we generated ISO-seq data for mixed RNA of five different tissues of Mo17, including root, silk, tassel and bract collected in the silking stage, and 14-d seedlings. All five tissues were frozen in liquid nitrogen after collecting and stored at −80°C before processing. Total RNA of each sample was extracted with TRIzol according to the manufacturer's instructions. The integrity of the RNA was determined with the Agilent 2100 Bioanalyzed (Agilent Technologies) and agarose gel electrophoresis. The purity and concentration of the RNA were determined with the Nanodrop (Thermo Fisher Scientific) and Qubit (Thermo Fisher Scientific). Only high-quality RNA (RNA integrity number ≥ 8, OD260/280 = 1.8–2.2, OD260/230 ≥ 2.0) was used for library construction. Equal amounts of RNA from the five tissues were pooled together. Mixed RNA was reverse transcribed into cDNA using a SMARTer PCR cDNA Synthesis Kit (PacBio Biosciences). Double-stranded cDNA was then generated by PCR (PrimeSTAR GXL DNA polymerase). Resulted double-stranded cDNAs were DNA damage repaired, end repaired and ligated to sequencing adapters using SMRTbell Template Prep Kit 1.0 (Pacific Biosciences). The SMRTbell template was annealed to sequencing primer and bound to polymerase and sequenced on the PacBio Sequel platform using Sequel Binding Kit 3.0 (Pacific Biosciences).

## Gene annotation

Both ab initio prediction and evidence-based prediction were used to predict the protein-coding genes in the Mo17 genome. For evidence-based prediction, four different approaches were performed, including RNA sequencing (RNA-seq) based prediction, ISO-seq-based prediction, protein-based homology search and evidence-based MAKER prediction. The methods of gene annotation are detailly described in the Supplementary Methods.

## Homolog analysis of genes

Homology analyses were performed for genes annotated in the genomes of Mo17 and NAM founder lines using OrthoFinder[84] (v2.5.2) with default settings. Protein sequences of the longest transcripts of each gene in the Mo17 genome were used for analysis. If the protein of a gene in the Mo17 genome was classified into the same Orthogroup with at least one of the genes annotated in the NAM founder lines by OrthoFinder, the gene was considered to have a homolog gene in NAM founder lines. The protein sequences of centromeric genes in the Mo17 genome were aligned to the predicted protein sequences of *Sorghum bicolor* (NCBIv3) using protein basic local alignment search tool (BLASTP)[85] (v2.9.0). The homologs of centromeric genes were identified in sorghum with a threshold of $E$ value less than $1 \times 10^{-3}$.

## Identification of duplicated genes

To identify gene duplications, BLASTP[85] was used to calculate pairwise similarities for proteins encoding by the longest transcripts of each

gene with a threshold of $E$ value $< 1.0 \times 10^{-20}$. MCscanX[86] was then used for classification with default parameters.

## Identification of nearly identical segmental duplications

The T2T Mo17 genome was aligned itself with Mummer[87]. Then, the region pairs with identity over 0.99 and length over 1,000 bp were identified as nearly identical segmental duplications from Mummer results. Overlapped and adjacent nearly identical segmental duplications were merged.

## Identification of satellite arrays

Satellite arrays were identified by BLAST with five reported satellite repeats (knob180 (ref. 88), TR-1 (ref. 47), CentC[89], Cent4 (ref. 48) and tRNASAT_ZM[49]) and three newly identified satellite repeats (sat268, sat261 and sat112) against the Mo17 genome. The sequences of knob180, TR-1 and CentC included in the MTEC (https://github.com/oushujun/MTEC/blob/master/maizeTE02052020) and the sequences of Cent4 and tRNASAT_ZM included in Repbase[49] were used for identification of corresponding satellite arrays. The sequences of repeat units of the three new satellite repeats identified are provided in Supplementary Note. The sequences of repeat units of the three new satellite repeats identified are provided in Supplementary Note. We split the genome into bins with 100 kb window size and 10 kb step size. Then we calculated the proportion of the eight satellites in each bin. For each of the eight satellites, the bins with more than 10% sequences for corresponding satellites were defined as arrays. Overlapped arrays with the same types of satellites were merged. Then the boundaries of merged arrays were trimmed to remove nonsatellite sequences at their ends.

## Higher-order repeat analysis

A position probability matrix (PPM) was generated for knob180, TR-1 and CentC repeats, respectively, and then a variant distance to the PPM was calculated for each satellite repeat copy according to the method reported previously[18]. Briefly, all entire copies of analyzed satellites in the genome were used for multiple sequence alignment using MAFFT[90,91] (--sparsescore 1000 --inputorder). The nucleotide frequencies at each alignment position were calculated to generate a PPM. Then, a 'variant distance' was calculated for each monomer by summation of disagreeing nucleotide probabilities (one minus corresponding nucleotide frequency) at each position via comparing with the PPM. For each of the three types of satellites, each monomer was compared to all other copies. The monomers with pairwise variant scores of 5 or less were clustered into the same group, which was termed as higher-order repeat group. A given monomer can be included for different higher-order repeat groups.

## Genotypes analysis of 5S and 45S rDNAs

The sequences of each intact copy of 5S and 45S rDNA in the final T2T assembly of the Mo17 genome were extracted. Then, multiple alignment was performed for the 5S rDNAs and 45S rDNAs, respectively, using MAFFT[90,91] (v7.475) with default parameters. For the 5S rDNAs, the SNPs and indels that were supported by more than 10% of 5S rDNA copies were selected and used for genotype analysis. For the 45S rDNAs, only the indels that were larger than 5 bp and were supported by more than 10% of 45S rDNA copies were selected and used for genotype analysis. For 45S rDNAs, all the indels selected were found to be located in the intergenic spacer region.

## Identification of centromeres

The anti-CENH3 chromatin immunoprecipitation sequencing (ChIP–seq) libraries were constructed using rabbit polyclonal anti-CENH3 against the peptide RPGTVALREIRKYQKSSTSATPERAAGTGGR, which was produced and supplied by GL Biochem and was the same as the anti-CENH3 used previously[92]. About 10 g of fresh Mo17 leaves were collected and used for nuclei extraction. Extracted nuclei were digested with micrococcal nuclease (Sigma, N3755), and then used for ChIP using the anti-CENH3. About 5 µl antibodies, with a concentration of 0.83 mg ml$^{-10}$, were used for per 25 µg chromatin. Two biological replicates were set. The ChIP–seq libraries were sequenced to generate 150-nucleotide paired-end reads on the Illumina HiSeq platform (Illumina). Resulting ChIP–seq data of CENH3 was used for the identification of functional centromere regions of the Mo17 genome. Raw reads were subjected to adapter trimming and filtering by fastp (v0.20.0) with the following parameters: '--dedup --detect_adapter_for_pe --cut_front --cut_tail'. Resulting ChIP–seq reads were mapped onto the Mo17 genome using Bowtie2 (ref. 93; v2.4.4) with the parameters of '--very-sensitive --no-mixed --no-discordant -k 10 -t -q -X 1000 -L 25'. Uniquely mapped reads were extracted by SAMTools[94] (parameters: -F 1804 -f 2 -q 20 -e '[NM] <= 2') for further analysis. Enrichment level of CENH3 for each base was obtained using bamCompare in the Deeptools package[95] (v3.5.1) with the parameters of '--binSize 1 --numberOfProcessors 40 --operation ratio --outFileFormat bedgraph'. Average enrichment of each 1 kb-bin of the genome was then calculated. The bins with enrichment levels greater than 5 were retained and with a distance interval less than 1 Mb were merged. The final centromeric regions were determined by visual inspection of the distribution of CENH3 ChIP–seq peaks.

## Identification of subtelomeres

Using RepeatMasker[82] (v4.1.1), the subtelomeric repeat sequences in the Mo17 genome were identified based on two previously reported subtelomeric repeats, including subtelomere sequence U39642.1 (ref. 56; GeneBank accession number U39642) and subtelomere sequence 4-12-1 (ref. 57; GeneBank accession no. CL569186.1) at first. Notably, we found that the subtelomeric repeats on the Mo17 genome were not completely consistent with the sequences of U39642.1 and 4-12-1. For subtelomeres identified by U39642.1, we found the corresponding repeat unit was a 629 bp sequence in the Mo17 genome, which was homologous with the sequences of U39642.1. This 629 bp sequence was termed subtelomeric repeat 1. In addition, we found 4-12-1 sequence was also not well matched with corresponding subtelomeric sequences identified in the Mo17 genome. Only part of 4-12-1 sequences could be matched to the corresponding subtelomeres identified, and 4-12-1 could not account for the whole length of the repeat unit of corresponding subtelomeres. Therefore, we used TRF[83] to redefine the subtelomeric repeat unit with the following parameters: 'trf 2 7 7 80 10 50 1000 -f -h -d -m', and got a 532 bp repeat unit, which was termed as subtelomeric repeat 2. A total of 22.58% of sequences of subtelomeric repeat 2 were homologous with the subtelomeric sequence 4-12-1. Next, based on the subtelomeric repeat 1 and 2, we reidentified the subtemomeric repeat sequences of the Mo17 genome using RepeatMasker[82]. The final subtelomeric regions for each end of ten chromosomes were determined by visual inspection of the distribution of subtemomeric repeat sequences identified. The sequences of the two subtelomeric repeats are provided in Supplementary Note.

## Reporting summary

Further information on research design is available in the Nature Portfolio Reporting Summary linked to this article.

## Data availability

The genome assembly and raw sequencing data generated in this study, including PacBio HiFi data, common ONT data, ultralong ONT data, ISO-seq data and ChIP–seq data, can be achieved from NCBI with BioProject number PRJNA751841. The GenBank accession number of the above data is JAIRCI000000000. The .fast5 format files of the ultralong ONT reads have been deposited in the National Genomics Data Center (NGDC), Beijing Institute of Genomics, Chinese Academy of Sciences, under BioProject accession number PRJCA012690. Genome assembly and gene annotation files can also be found in

CyVerse (https://data.cyverse.org/dav-anon/iplant/home/laijs/Zm-Mo17-REFERENCE-CAU-2.0/). The Illumina PCR-free data used in this study can be obtained from NCBI under accession number SRP111315 (ref. 13). The RNA-seq data used for gene annotation can be achieved from NCBI under accession numbers GSE16916 (ref. 8), GSE54272 (ref. 96), GSE57337 (ref. 97), GSE61810 (ref. 98), GSE70192 (ref. 99), GSE43142 (ref. 100), SRP051572 (ref. 101), SRP064910 (ref. 102), SRP052226 (ref. 103), SRP006703 (ref. 104), SRP009313 (ref. 105), SRP010124 (ref.106), SRP011187 (ref.107), SRP011480 (ref.108), SRP013432 (ref.109), SRP015339 (ref. 110), SRP110782 (ref. 111), SRP111315 (ref. 13), SRP017111 (ref. 112), SRP018088 (ref. 113), SRP026161 (ref. 114) and SRP029742 (ref. 115). The detail runs of published RNA-seq data used are demonstrated in Supplementary Table 14.

## Code availability
Custom scripts and codes used in this study are available at GitHub (https://github.com/LAILAB-CAU/update-Mo17) and Zenodo[116] (https://doi.org/10.5281/zenodo.7833112).

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

## Acknowledgements

This study is dedicated to the memory of J. Messing (1946–2019). We thank F. Han (Institute of Genetics and Developmental Biology, Chinese Academy of Sciences) and Y. Liu (Institute of Genetics and Developmental Biology, Chinese Academy of Sciences) for their help in the construction of the anti-CENH3 ChIP–seq libraries. This work was supported by grants from the National Key Research and Development Program of China (2021YFF1000500), the Hainan Yazhou Bay Seed Lab (B21HJ0509), the Henan Modern Seed Industry Co. (2022010202-3), the Yazhou Bay Science and Technology City Administration (SYND-2022-03) and the National Natural Science Foundation of China (62031003).

## Author contributions

J.L. conceived the project. J.C. and Z.W. performed genome assembly and analysis of repeat sequences. J.C., Z.W. and B.X. performed assembly quality evaluation. K.T. and Z.W. performed genome annotation. T.L. performed analysis of CENH3 ChIP–seq data. W.H. and W.J. performed FISH assay. J.S., J.H., K.W. and C.W. were involved in the generation of the ONT and PacBio data, as well as initial ONT and PacBio assemblies. Z.W. W.S. and H.Z. collected the tissues used for ONT and PacBio sequences. M.B.H. and J.C.S. participated in the discussion of the results. J.C. and J.L. wrote the manuscript.

## Competing interests

The authors declare no competing interests.

## Additional information

**Extended data** is available for this paper at https://doi.org/10.1038/s41588-023-01419-6.

**Correspondence and requests for materials** should be addressed to Jinsheng Lai.

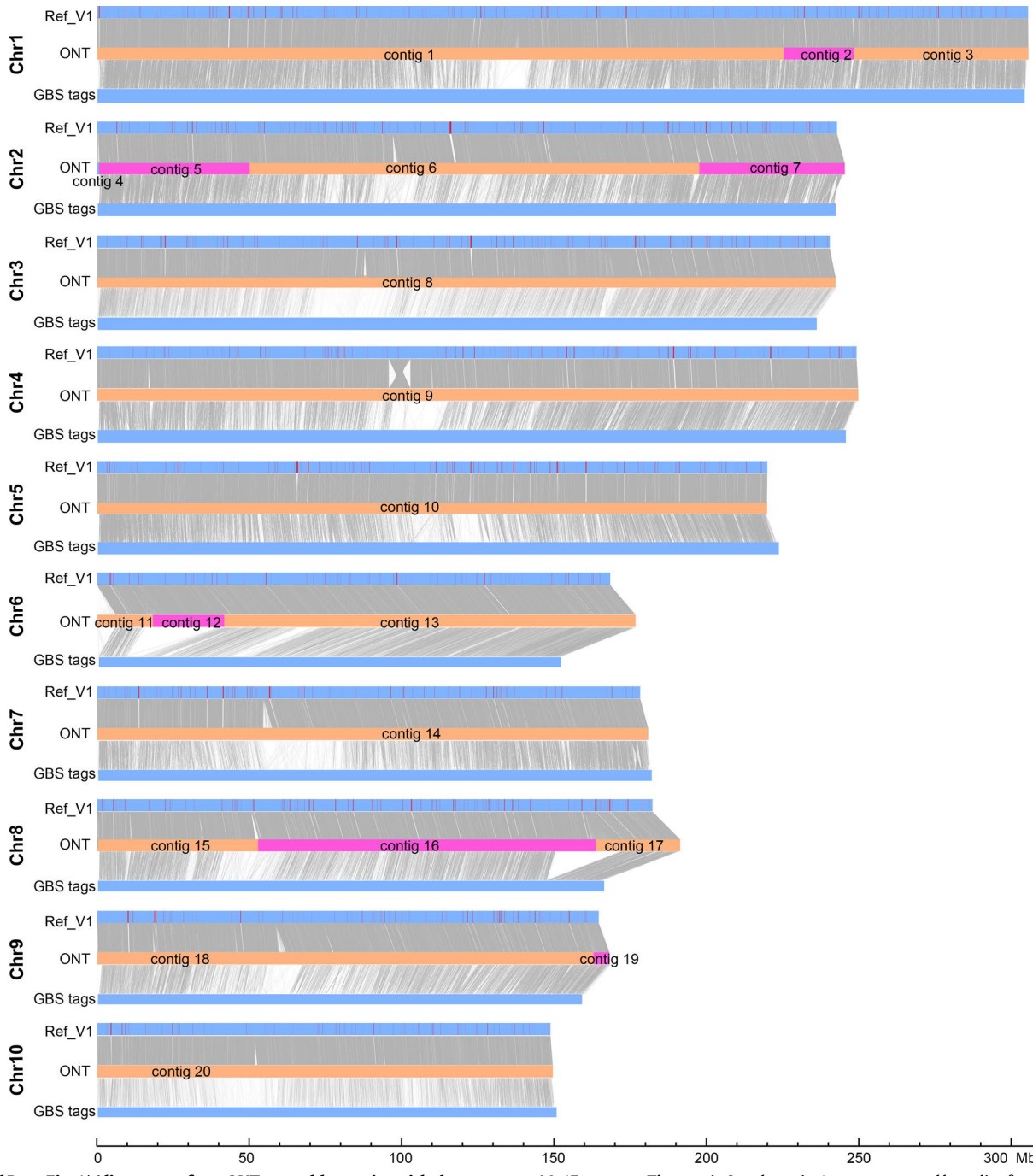

**Extended Data Fig. 1 | Alignment of raw ONT assembly contigs with the pseudomolecules of the Mo17ref_V1 and the GBS tags.** There was a total of 20 contigs which were anchored and oriented onto ten pseudomolecules of the Mo17 genome. The contig 3 and contig 6 were generated by split of a raw ONT contig with assembly error (see Supplementary Fig. 1). The red lines on the blue blocks refer the gaps on the Mo17ref_V1.

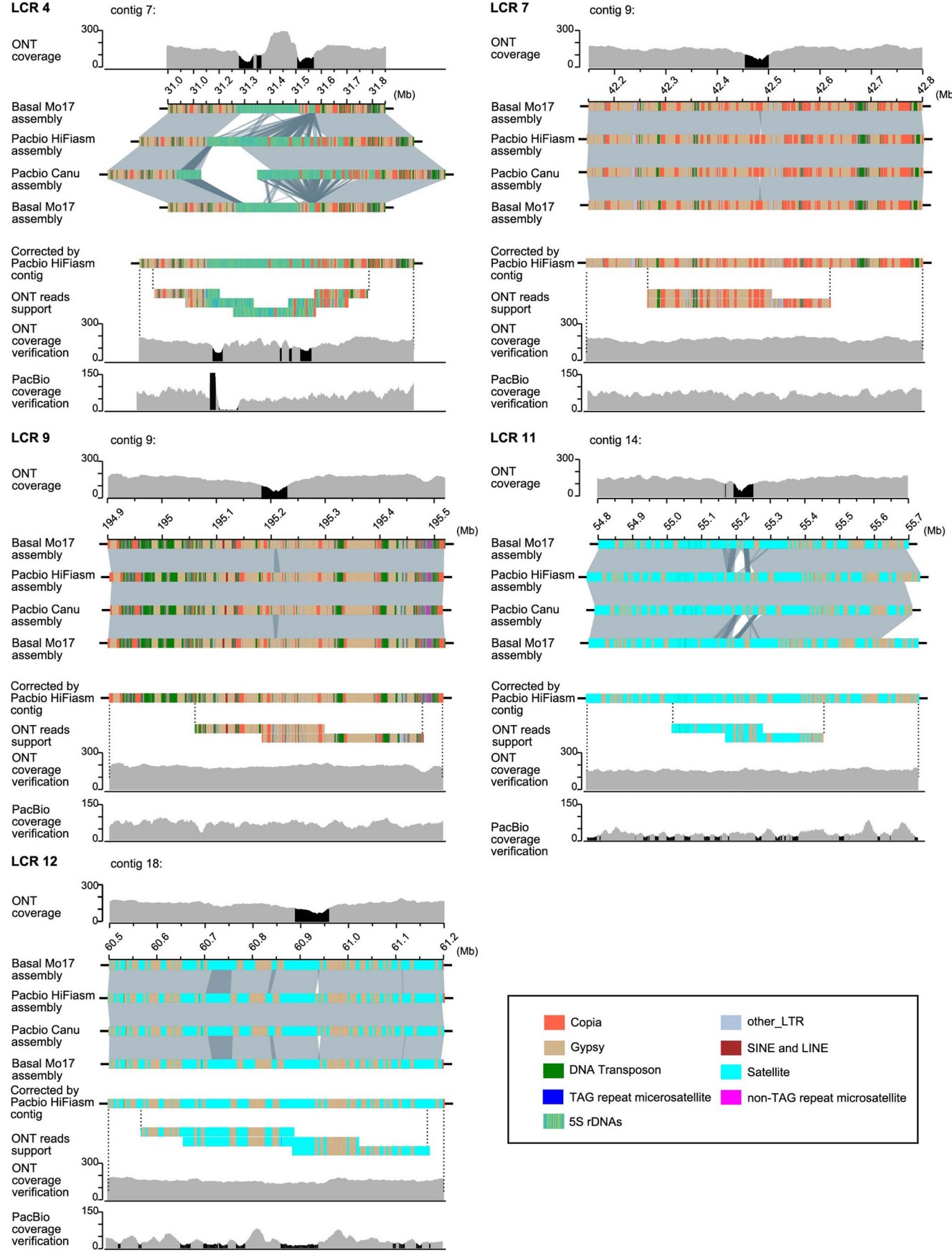

**Extended Data Fig. 2 | Correction of five LCRs by the contigs of PacBio assembly.** According to the alignment with the PacBio assembly, 5 LCRs on the basal Mo17 assembly could be corrected by the contigs of PacBio Hifiasm assembly. Corrected assembly was confirmed by the uniform ONT and/or PacBio reads coverage, and tiling ONT reads. Black shades refer local coverage-anomalous regions.

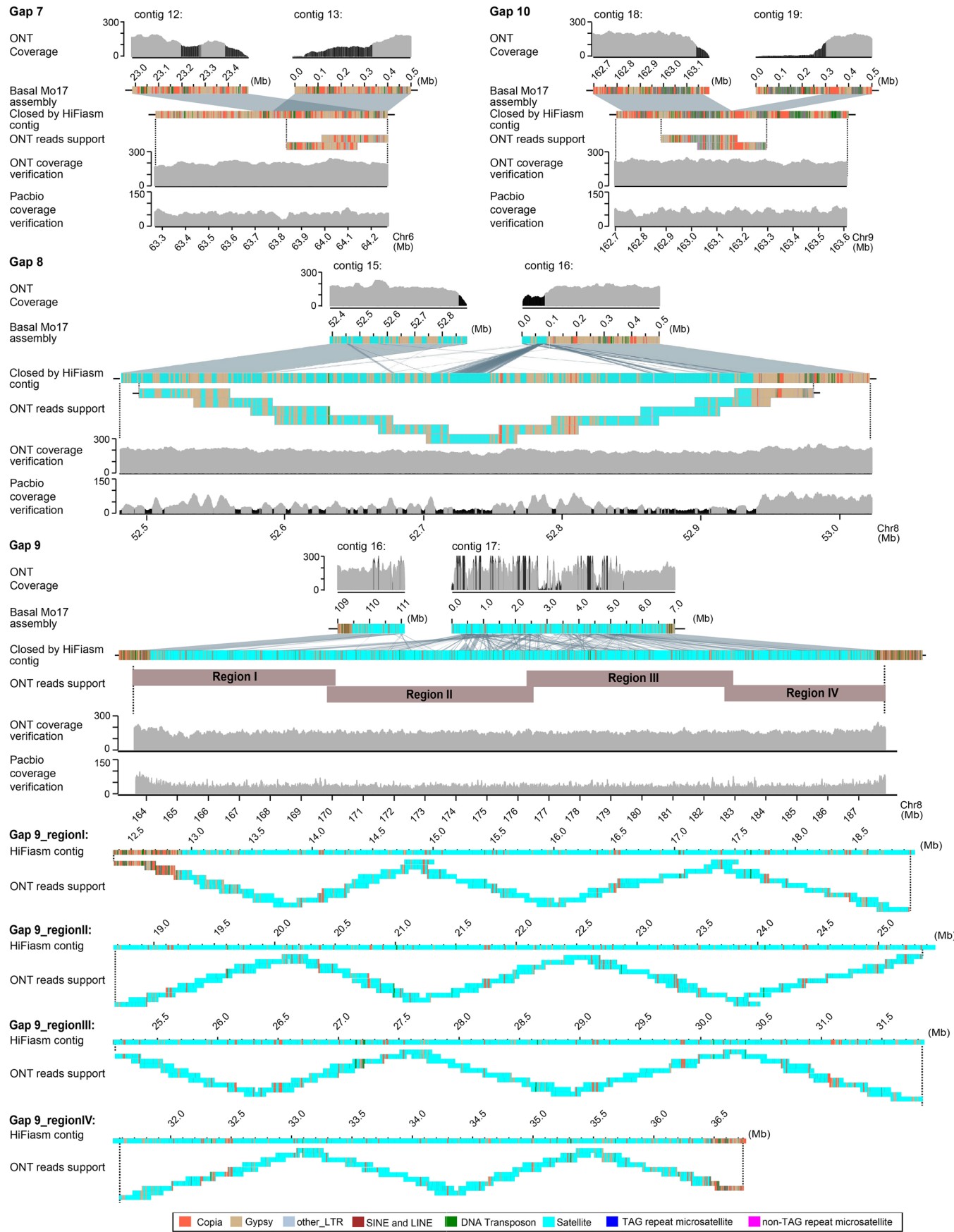

**Extended Data Fig. 3 | See next page for caption.**

**Extended Data Fig. 3 | Validation the assembly of gaps closed by the contigs of PacBio assembly.** The validity of four gaps closed by the contigs of PacBio Hifiasm assembly were confirmed by the uniform ONT and/or PacBio reads coverage and tiling ONT reads. Black shades refer local coverage-anomalous regions. We noted that in gaps 7 and 10, a part of the ONT contigs not aligned to HiFi contigs were removed in the final assembly. Compared to the normal gap ends (which were remained in the final assembly) with gradually decreased coverage, almost no aligned ONT reads were observed for these regions removed in the final assembly, which suggested that these removed parts of contigs were in fact redundant or misassembled fragments, and further confirmed the assemblies of these gap-closed regions.

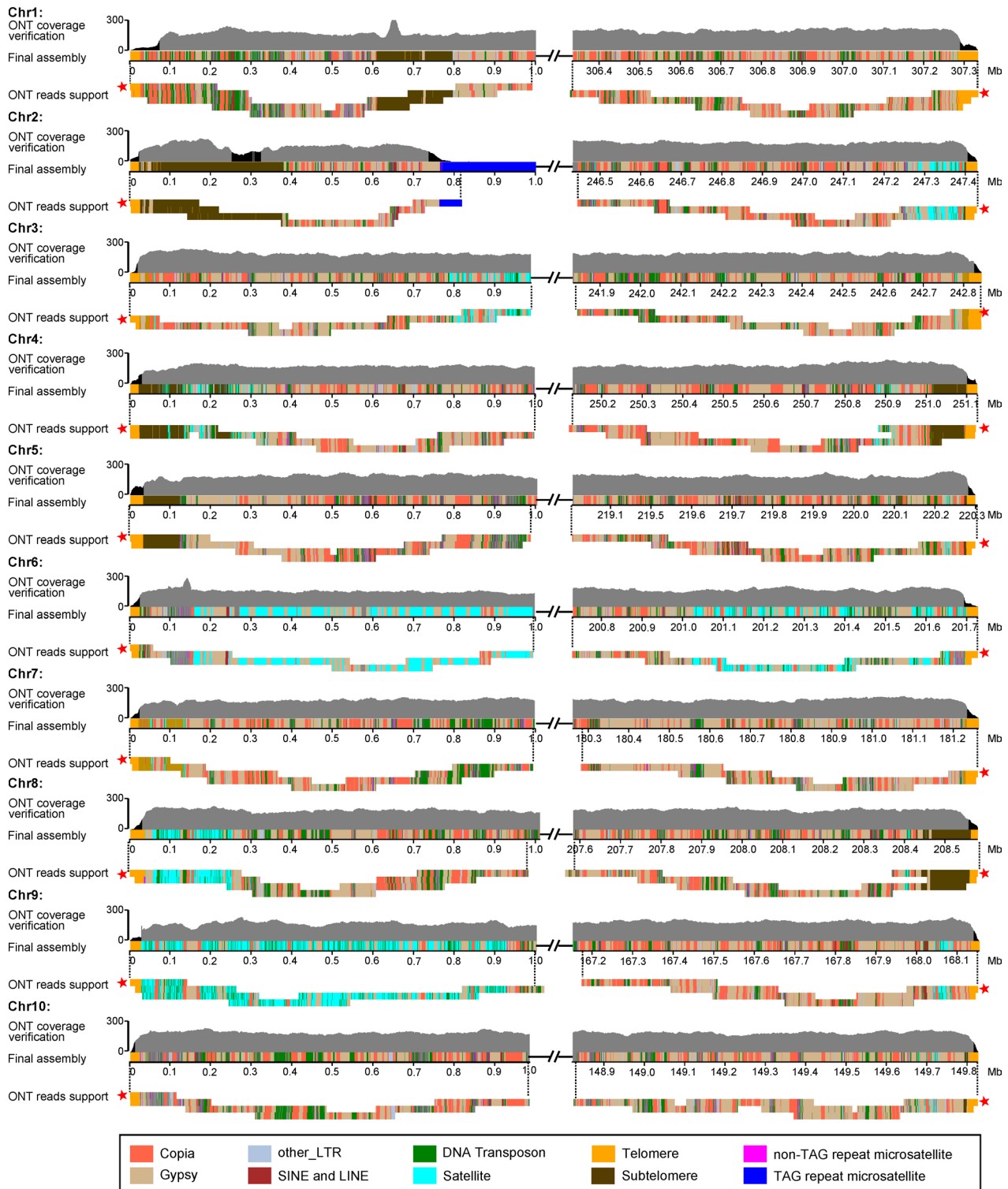

**Extended Data Fig. 4 | Validation of the assembly of the terminal 1 Mb regions for 10 chromosomes of the final Mo17 assembly.** The assembly of the terminal 1 Mb regions of all 10 chromosomes were confirmed by tiling ONT reads. ONT reads coverage analysis showed that expect for the 20 telomeres of chromosomes, as well as subtelomeric regions and long TAG repeat region on chromosome 2, uniform ONT reads coverage were observed in general. Black shades of ONT coverage verification refer the regions with reads depth lower than 100. Red pentagrams refer the reads harbored with the longest telomeric repeats for corresponding chromosomal ends, which were used to correct corresponding telomeric regions.

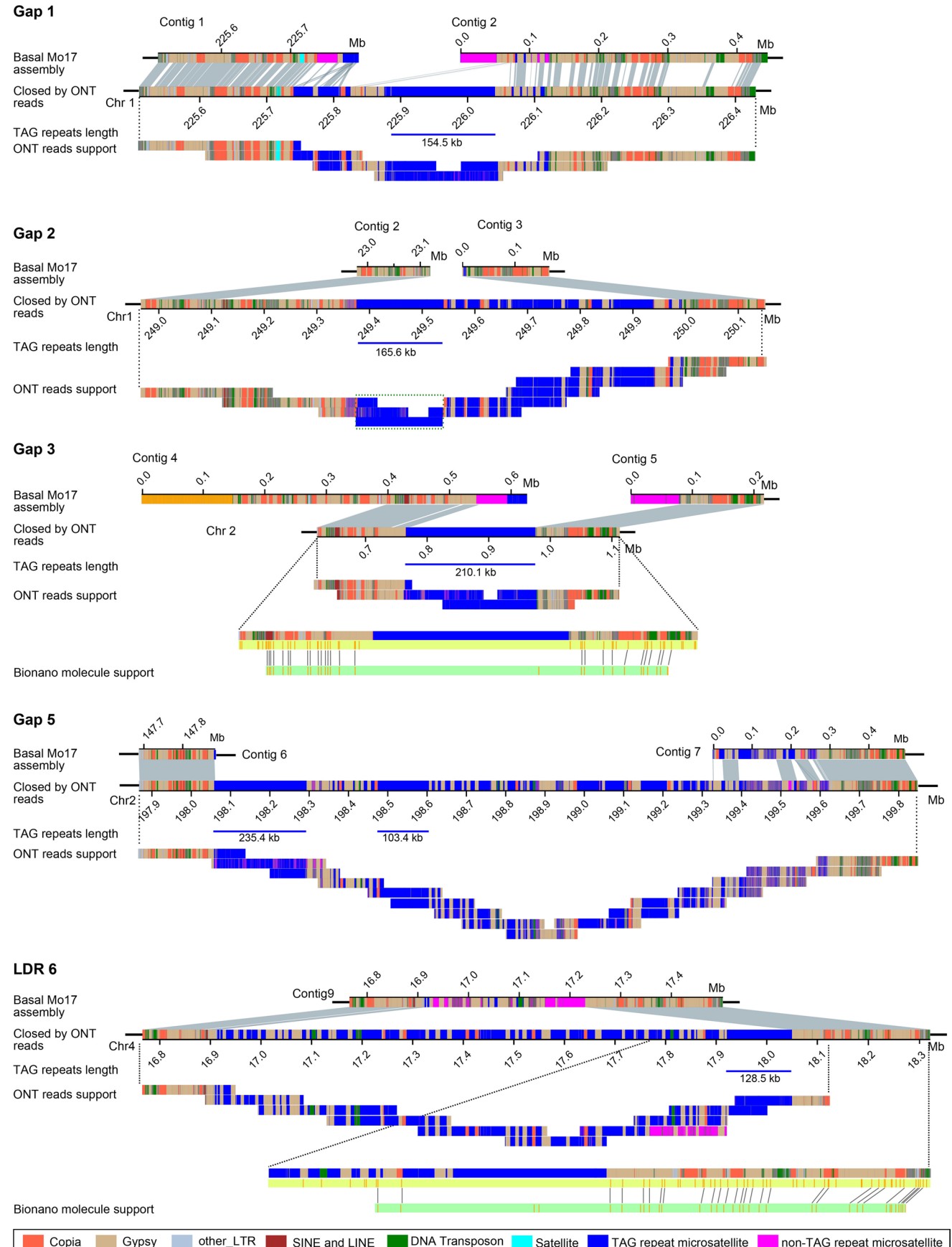

**Extended Data Fig. 5 | Schematic showing the manual closing for 5 TAG repeat array related gaps by ONT reads.** The green dotted box indicated the TAG repeat region in gap2 which length was estimated (see Method).

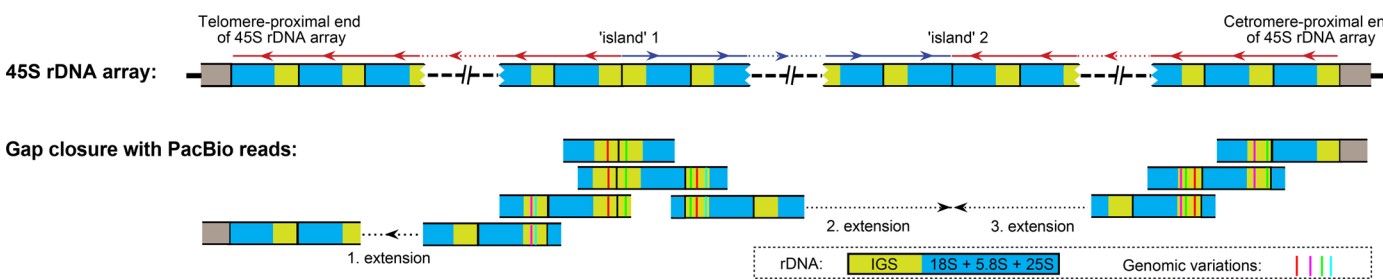

**Extended Data Fig. 6 | Schematic of 45S rDNA related gap closure by PacBio HiFi reads.** The blue and red arrows indicated the transcription directions of corresponding 45S rDNAs were toward to the centromere and telomere, respectively. The black arrows indicated the direction of extension during gap closure.

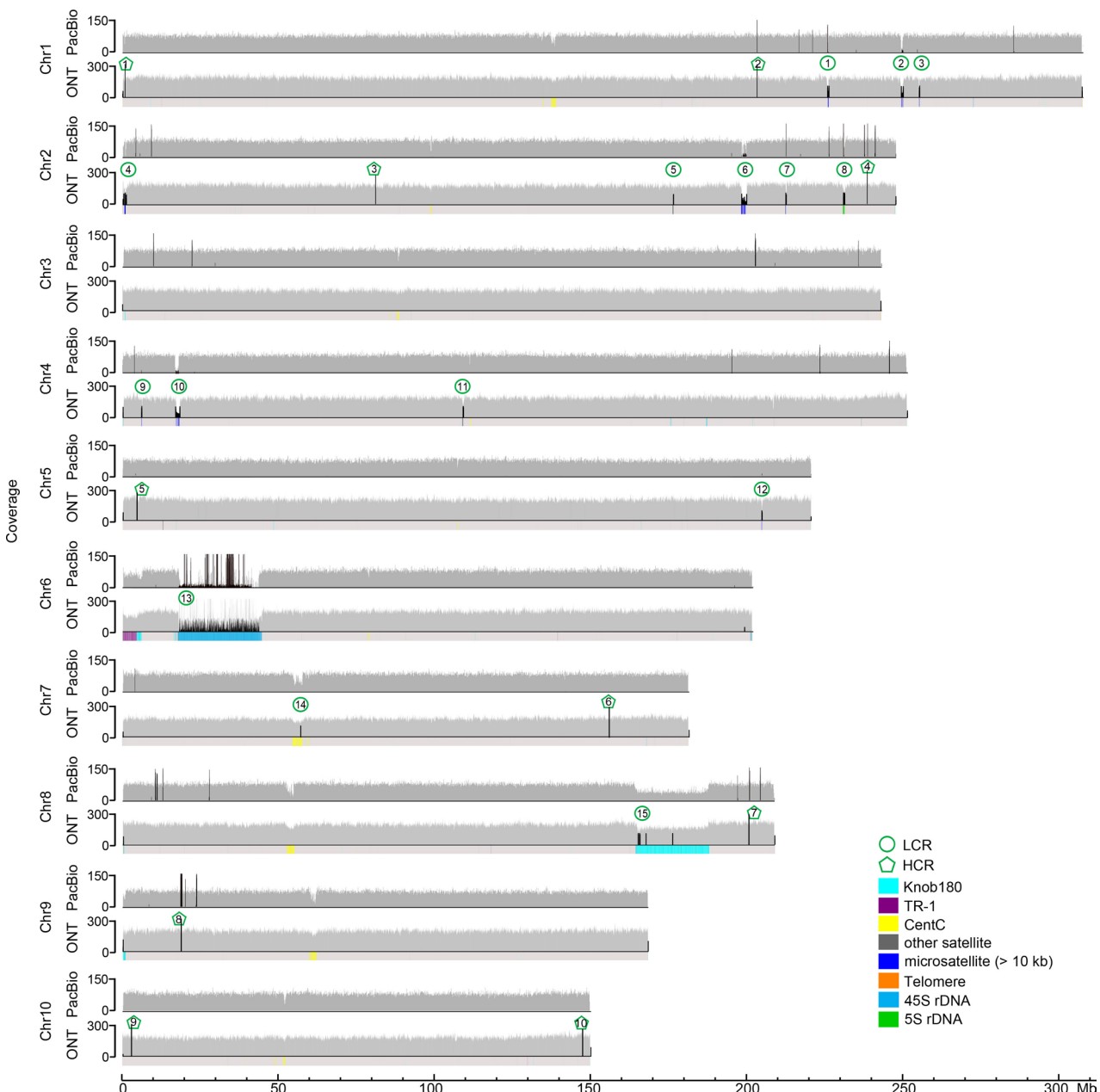

**Extended Data Fig. 7 | Whole-genome coverage of ONT reads across the T2T assembly of Mo17 genome.** Ultra-long ONT reads longer than 10 kb and PacBio HiFi reads were used for analysis. Local coverage-anomalous regions were shown in black shades.

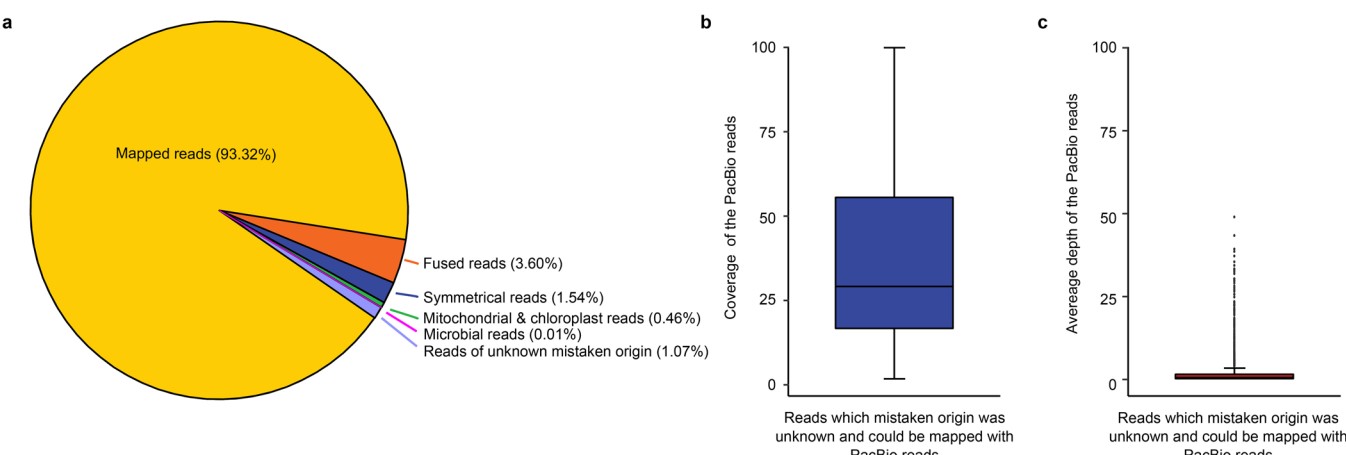

**Extended Data Fig. 8 | Validation of the completeness of the T2T Mo17 assembly by mapping of ONT reads. a**, The composition of different types of ONT reads. Quality filtered ONT reads that longer than 10 kb were used for analysis. Reads of unknown mistaken origin refer the unmapped reads which were not grouped as fused reads, symmetrical reads, microbial reads, and mitochondrial and chloroplast reads based on the thresholds we used (see Method). **b, c,** Box plots showing the coverage (b) and average depth (c) of PacBio reads across the reads (n = 9,518) which mistaken origin was unknown and could be mapped with PacBio reads. In box plots, the 25% and 75% quartiles are shown as lower and upper edges of boxes, respectively, and central lines denote the median. The whiskers extend to 1.5 times the interquartile range. Data beyond the end of the whiskers are displayed as outlying dots. Totally, there were 83,167 reads of unknown mistaken origin, which nearly 85% could not be supported by any of PacBio reads. There were only 15% (13,636/83,167) reads of unknown mistaken origin could be mapped with PacBio reads. However, average only 38.1% regions of these 13,636 reads of unknown mistaken origin were covered by PacBio reads. In addition, average depth of PacBio reads across these 13,636 reads of unknown mistaken origin were only 1.8×, respectively, far lower than the theoretical 69.4×. Consequently, no any reliable PacBio reads supports were found for reads of unknown mistaken origin.

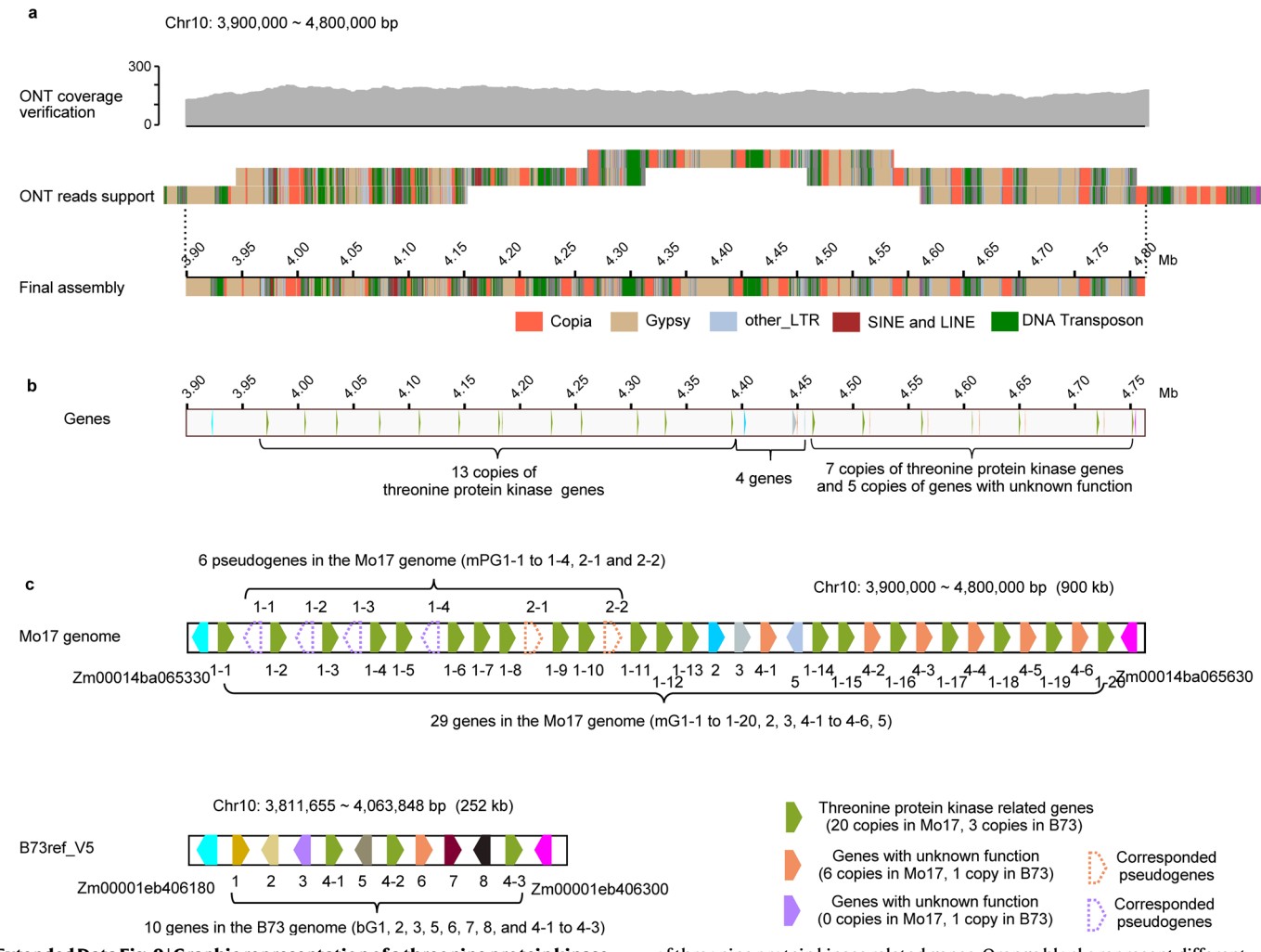

**Extended Data Fig. 9 | Graphic representation of a threonine protein kinase related tandem gene loci on chromosome 10. a**, The validity of the assembly of corresponding region on the Mo17 genome was confirmed by uniform ONT reads coverage and tiling ONT reads. **b**, The locations of genes annotated in corresponding region of Mo17 genome. Green blocks represent different copies of threonine protein kinase related genes. Orange blocks represent different copies of another duplicated gene with unknown function. **c**, Schematic showing the genes and pseudogenes in corresponding regions of the B73 and Mo17 genomes.

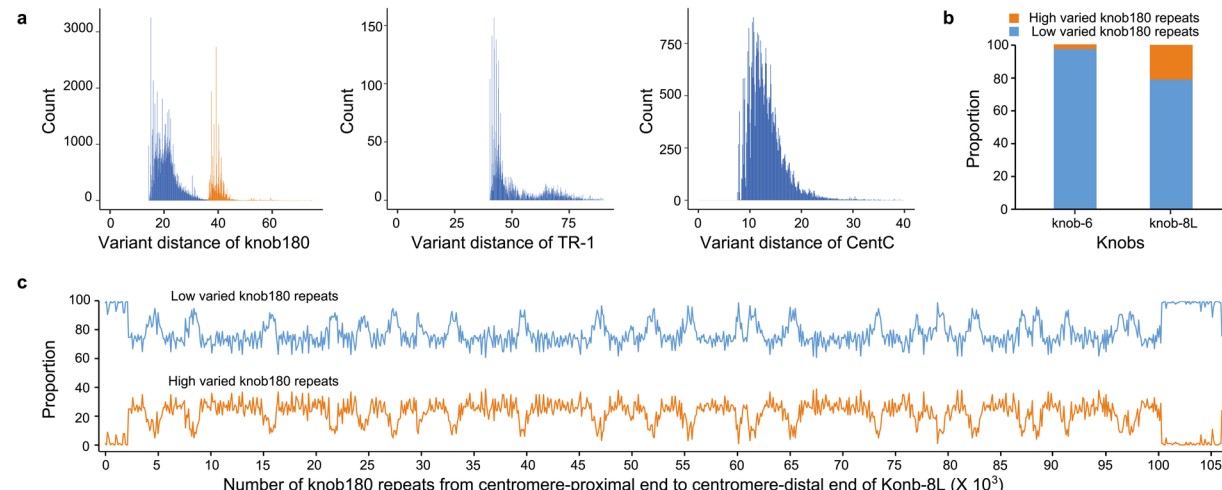

**Extended Data Fig. 10 | Variant distances of knob180, TR-1, and CentC repeats. a**, Histograms of the variant distances relative to the genome-wide consensus for knob180, TR-1, and CentC repeats. **b**, Proportion of knob180 repeats with relatively low and high levels of variant distances on Knob-6S and Knob-8L. **c**, The distribution of knob180 repeats with relatively low and high levels of variant distances along Konb-8L.

# Reporting Summary

## Statistics

For all statistical analyses, confirm that the following items are present in the figure legend, table legend, main text, or Methods section.

| n/a | Confirmed | |
|---|---|---|
| ☐ | ☒ | The exact sample size ($n$) for each experimental group/condition, given as a discrete number and unit of measurement |
| ☐ | ☒ | A statement on whether measurements were taken from distinct samples or whether the same sample was measured repeatedly |
| ☐ | ☒ | The statistical test(s) used AND whether they are one- or two-sided *Only common tests should be described solely by name; describe more complex techniques in the Methods section.* |
| ☒ | ☐ | A description of all covariates tested |
| ☐ | ☒ | A description of any assumptions or corrections, such as tests of normality and adjustment for multiple comparisons |
| ☐ | ☒ | A full description of the statistical parameters including central tendency (e.g. means) or other basic estimates (e.g. regression coefficient) AND variation (e.g. standard deviation) or associated estimates of uncertainty (e.g. confidence intervals) |
| ☐ | ☒ | For null hypothesis testing, the test statistic (e.g. $F$, $t$, $r$) with confidence intervals, effect sizes, degrees of freedom and $P$ value noted *Give P values as exact values whenever suitable.* |
| ☒ | ☐ | For Bayesian analysis, information on the choice of priors and Markov chain Monte Carlo settings |
| ☒ | ☐ | For hierarchical and complex designs, identification of the appropriate level for tests and full reporting of outcomes |
| ☐ | ☒ | Estimates of effect sizes (e.g. Cohen's $d$, Pearson's $r$), indicating how they were calculated |

*Our web collection on statistics for biologists contains articles on many of the points above.*

## Software and code

Policy information about availability of computer code

| Data collection | PacBio SequeII platform produced HiFi SMRT long reads. Nanopore PromethION platform produced ONT common and ultra-long reads. PacBio Sequel platform produced ISO-seq reads. Illumina HiSeq platform produced ChIP-seq reads. |
|---|---|
| Data analysis | Genome assembly: NextDenovo v2.2-beta.0, NextPolish v1.1.0, Hifiasm v.0.7, Canu v.2.0; Gap closure: Minimap2 v2.17, BLASTN v2.9.0, Solve v3.5.1; Satellite analysis: MAFFT v7.475; Genome evaluation: Merqury v1.1, BWA v0.7.17; Genome annotation: EDTA v1.7.0, RepeatMasker v4.1.1, trf v4.09 , Fgenesh v7.2.2, STAR v2.7.8a, FASTP v0.20.0, StringTie v2.1.2, Cufflinks v2.2.1, CLASS2 v2.1.7, TACO v0.7.3, Mikado v2.0rc2, SAMTools v1.9, TransDecoder v5.5.0, Diamond v2.0.1, PASA v2.3.3, GMAP v.2017-11-15, MMseqs v12.113e3, GeMoMa v1.6.4, MAKER v 2.31.10, InterProScan v5.39-77.0; Centromere identification: Bowtie2 v2.4.4, Deeptools v3.5.1, BEDTools v2.29.2; Identification of duplicated genes: McscanX(https://github.com/wyp1125/MCScanX), OrthoFinder v2.5.2; Customized scripts in GitHub repository: https://github.com/LAILAB-CAU/update-Mo17. |

For manuscripts utilizing custom algorithms or software that are central to the research but not yet described in published literature, software must be made available to editors and reviewers. We strongly encourage code deposition in a community repository (e.g. GitHub). See the Nature Portfolio guidelines for submitting code & software for further information.

## Data

Policy information about availability of data

All manuscripts must include a data availability statement. This statement should provide the following information, where applicable:

- Accession codes, unique identifiers, or web links for publicly available datasets
- A description of any restrictions on data availability
- For clinical datasets or third party data, please ensure that the statement adheres to our policy

The genome assembly and raw sequencing data generated in this study, including PacBio HiFi data, common ONT data, ultra-long ONT data, ISO-seq data, and ChIP-seq data can be achieved from NCBI with BioProject number PRJNA751841. The GenBank accession number of the above data is JAIRCI000000000. The .fast5 format files of the ultra-long ONT reads have been deposited in the National Genomics Data Center (NGDC), Beijing Institute of Genomics, Chinese Academy of Sciences, under BioProject accession number PRJCA012690. Genome assembly and gene annotation files can also be found in CyVerse(https://data.cyverse.org/dav-anon/iplant/home/laijs/Zm-Mo17-REFERENCE-CAU-2.0/). The Illumina PCR-free data used in this study can be obtained from NCBI under accession number SRP111315. The RNA-seq data used for gene annotation can be achieved from NCBI under accession numbers of GSE16916, GSE54272, GSE57337, GSE61810, GSE70192, GSE43142, SRP051572, SRP064910, SRP052226, SRP006703, SRP009313, SRP010124, SRP011187, SRP011480, SRP013432, SRP015339, SRP110782, SRP111315, SRP017111, SRP018088, SRP026161, and SRP029742. The detail runs of published RNA-seq data used are demonstrated in Supplementary Table 14.

# Field-specific reporting

Please select the one below that is the best fit for your research. If you are not sure, read the appropriate sections before making your selection.

☒ Life sciences          ☐ Behavioural & social sciences          ☐ Ecological, evolutionary & environmental sciences

For a reference copy of the document with all sections, see nature.com/documents/nr-reporting-summary-flat.pdf

# Life sciences study design

All studies must disclose on these points even when the disclosure is negative.

| | |
|---|---|
| Sample size | The sample size for genome assembly was the number of inbred lines. We choose only one maize inbred lines (Mo17) for sequencing and assembly. The sample size for the statistic analysis of each experiment was clearly mentioned in each figure legend or Methods. |
| Data exclusions | Only quality filtered ultra-long ONT reads were used for ONT data based assembly. |
| Replication | Replications for each experiment were clearly stated in figure legends or Methods section. Replications of repeated experiments were evaluated by proper statistic analyses and confirmed to be successful. |
| Randomization | For each individual of Mo17 inbred line, the sampling process for DNA sequencing, ISO-seq and ChIP-seq was randomly conducted. |
| Blinding | Blinding is not necessary for genome sequencing and assembly, since the investigators know which maize species they were handling. No blinding should not affect interpretation as all our experiment measures were objective. |

# Reporting for specific materials, systems and methods

We require information from authors about some types of materials, experimental systems and methods used in many studies. Here, indicate whether each material, system or method listed is relevant to your study. If you are not sure if a list item applies to your research, read the appropriate section before selecting a response.

## Materials & experimental systems

| n/a | Involved in the study |
|---|---|
| ☐ | ☒ Antibodies |
| ☒ | ☐ Eukaryotic cell lines |
| ☒ | ☐ Palaeontology and archaeology |
| ☒ | ☐ Animals and other organisms |
| ☒ | ☐ Human research participants |
| ☒ | ☐ Clinical data |
| ☒ | ☐ Dual use research of concern |

## Methods

| n/a | Involved in the study |
|---|---|
| ☐ | ☒ ChIP-seq |
| ☒ | ☐ Flow cytometry |
| ☒ | ☐ MRI-based neuroimaging |

## Antibodies

| | |
|---|---|
| Antibodies used | CENH3 antibody. The CENH3 antibody is a rabbit polyclonal against the peptide RPGTVALREIRKYQKSSTSATPERAAGTGGR. The |

| Antibodies used | antibodies were custom-produced and supplied by GL Biochem. |
|---|---|
| Validation | The rabbit polyclonal antibody against CENH3 was validated to be available according to a previous report (Fu, Shulan, et al. "De novo centromere formation on a chromosome fragment in maize." Proceedings of the National Academy of Sciences 110.15 (2013): 6033-6036), in which the anti-CENH3 antibody used is same as we used here. |

# ChIP-seq

## Data deposition

☒ Confirm that both raw and final processed data have been deposited in a public database such as GEO.

☒ Confirm that you have deposited or provided access to graph files (e.g. BED files) for the called peaks.

| Data access links<br>*May remain private before publication.* | https://www.ncbi.nlm.nih.gov/bioproject/?term=PRJNA751841 |
|---|---|
| Files in database submission | SRR21509776, SRR21509777, SRR21509778, SRR21509779 |
| Genome browser session<br>(e.g. UCSC) | no longer applicable |

## Methodology

| Replicates | Two biological replicates were set |
|---|---|
| Sequencing depth | Depth 10 X; total reads,  158,687,990;  mapped reads  58,605,672, unique mapped reads  27,806,396; 150 bp pared-end reads |
| Antibodies | anti-CENH3 |
| Peak calling parameters | Enrichment level of CENH3 for each base was obtained using bamCompare in the Deeptools packag (v3.5.1) with the parameters of '--binSize 1 --numberOfProcessors 40 --operation ratio --outFileFormat bedgraph'. Average enrichment of each 1 kb-bin of the genome was then calculated. The bins that enrichment levels greater than 5 were retained, which with a distance interval less than 1 Mb were merged. The final centromeric regions were determined by visual inspection of the distribution of CENH3 ChIP-seq peaks. |
| Data quality | All ten centromeres in maize were successfully identified. |
| Software | Enrichment level of CENH3 for each base was obtained using bamCompare in the Deeptools package (v3.5.1). |

