## [Peer Review File · Nature Genetics]

Peer Review Information

Manuscript Title: A complete telomere-to-telomere assembly of the maize genome

Corresponding author name(s): Professor Jinsheng Lai

Reviewer Comments & Decisions:

Decision Letter, initial version:

26th Apr 2022

Dear Professor Lai,

Your Article, "A complete telomere-to-telomere assembly of the maize genome" has now been seen by 4 referees. You will see from their comments copied below that while they find your work of considerable potential interest, they have raised quite substantial concerns that must be addressed. In light of these comments, we cannot accept the manuscript for publication, but would be very interested in considering a revised version that addresses these serious concerns.

We hope you will find the referees' comments useful as you decide how to proceed. If you wish to submit a substantially revised manuscript, please bear in mind that we will be reluctant to approach the referees again in the absence of major revisions.

To guide the scope of the revisions, the editors discuss the referee reports in detail within the team, including with the chief editor, with a view to identifying key priorities that should be addressed in revision. In this case, we think all referees have provided constructive reviews aimed at strengthening the validation of the assembly and clarifying some methodologies, and we particularly ask that you sufficiently validate the structure of the assembly and address their technical comments as thoroughly as possible with appropriate revisions. We hope that you will find the prioritized set of referee points to be useful when revising your study. Please do not hesitate to get in touch if you would like to discuss these issues further.

If you choose to revise your manuscript taking into account all reviewer and editor comments, please highlight all changes in the manuscript text file. At this stage we will need you to upload a copy of the manuscript in MS Word .docx or similar editable format.

We are committed to providing a fair and constructive peer-review process. Do not hesitate to contact us if there are specific requests from the reviewers that you believe are technically impossible or

unlikely to yield a meaningful outcome.

*2) If you have not done so already please begin to revise your manuscript so that it conforms to our Article format instructions, available [here](http://www.nature.com/ng/authors/article_types/index.html). Refer also to any guidelines provided in this letter.

[redacted]

If you wish to submit a suitably revised manuscript we would hope to receive it within 6 months. If you cannot send it within this time, please let us know. We will be happy to consider your revision so long as nothing similar has been accepted for publication at Nature Genetics or published elsewhere. Should your manuscript be substantially delayed without notifying us in advance and your article is eventually published, the received date would be that of the revised, not the original, version.

Thank you for the opportunity to review your work.

Sincerely,
Wei

Wei Li, PhD
Senior Editor
Nature Genetics
New York, NY 10004, USA
www.nature.com/ng

Reviewers' Comments:

Reviewer #1:
Remarks to the Author:

This paper applies the latest long read sequencing and assembly methods to the maize genome, creating a 'T2T' map of the Mo17 strain. This does seem like a significant advance but the authors need to be more explicit about how this assembly is better than Hufford et al 2021 Science, which reported assemblies of 25 NAM founder inbreds and B73. Also recent B73 Jiao et al 2017 and Mo17 Sun et al 2018 assemblies. I particularly wasn't clear how different the assembly here is to the Mo17 2018 assembly - please spell this out.

Due to the relevance of maize to plant genetics and agriculture this is an important genome and makes advances, particularly assembling the first large rDNA clusters in plants (to my knowledge).

The authors generated large (237.7x) ONT and PacBio (69.4x) datasets. Lines 103 to 106 - please explain the difference between ONT 'raw ultra long' and 'common data' - what is the difference between the datasets?

I note the authors used Mo17ref_V1 to 'anchor and orientate' with - have the authors tried reference free methods of orientating/scaffolding? Could any possible errors in the V1 assembly be carried over at this step? Lines 120-123 - please define 'fairly uniform coverage' with respect to ONT read alignments to the assembly. Did the authors also try aligning PacBio HiFi reads to the assembly in a similar way? Please succinctly explain what the low coverage regions are in more detail (lines 122-123). Lines 129-130 - please provide more detail on how the PacBio contigs and ONT assembly were integrated - how congruent were the individual assemblies before integration? In the case of differences, how do you decide which alternative is correct?

Several recent papers, for example in human and Arabidopsis, have reported successful gapless assembly of centromere repeat arrays. However, the 5S and 45S rDNA have remained very challenging. The authors propose to have successfully assembled these arrays in maize. To validate these assemblies, a 'blast-based copy number estimation' is performed - please provide more explanation - I'm unclear how this provides validation? I'm also unclear why the authors think that k-mer based estimates from Illumina data would likely underestimate rDNA copy number? I can

appreciate that there would be many identical k-mers due to the low variation, but would this not be reflected in a higher copy number? In any case, as the authors have Illumina reads for this strain, it would be interesting to see how estimates using this method compared to the copy number estimates from the assembly. Line 455 – which TEs were found within the 5S – were they different from the surrounding regions?

The rDNA array structures shown in Fig 4D and 4E are very interesting and provide a potential first look inside complete plant rDNA clusters - the 'gradient' of TEs and rDNA variation seen the 5S cluster is particularly interesting.

One major suggestion I have for the paper, is that as the authors have abundant ONT data, and DeepSignal-plant has been proven to work well for plant repeat regions (eg Naish et al 2021), it would be beneficial to the study if DNA methylation maps across the T2T genome could be presented in this study. For example, plotting methylation on the very interesting rDNA assemblies in Fig 4D and 4E would be very interesting.

The paper also reveals large (up to 235 kp) tri-nucleotide repeats that are validated cytogenetically. Equally, I would be interested to know the methylation state of these repeats.

Line 131 – which transposon families are involved at these regions?

Line 153 – how do the authors know that ONT is introducing 'extra' sequence errors in TAG repeats?

Line 157 – I'm not clear how you know which reads are mistakenly read, and which are not?

An iterative alignment method using PacBio data is reported to assemble the 45S, ultimately building 2,974 copies. I'm unclear how the authors can be confident in this assembly although the read tiling path shown looks convincing.

Lines 304 – 307 – what is the cause of the high read coverage mapping here? This would seem to indicate missing sequences/copies from the assembly, causing a stack up of reads on the copies that are present. With respect to the mitochondrial insertions – recent work suggests that nuclear mtDNA sequences are likely to be DNA methylated, whereas the organelle copies will not be. Therefore, could the ONT reads be assessed for methyl state, as a means to restrict assembly to those most likely to represent nuclear insertions (assuming they are methylated). See <https://www.biorxiv.org/content/10.1101/2022.02.22.481460v1>

Line 158 – I'm surprised that this inversion has not been observed previously? For example, has crossover suppression been observed within this inversion in genetic maps made between Mo17 and strains that lack the inversion? Could it be possible that the inversion is polymorphic between Mo17 strains used for different genome assemblies?

For the telomeres, how do these size estimates compare to published work on the maize telomeres? Certainly in many plants telomere size has been estimated by Southern blotting (eg work of Shippen lab) – perhaps this data exists for maize and can be compared to?

Lines 338 – 342 – how many of these sequences are not represented in the previous Mo17 assembly? Were some expanded more than others in the new assembly relative to V1? Where do the new regions locate on the chromosomes? How many are centromeric? Line 344 – define 'colocalized' here more

clearly – what relationship are the authors proposing between the TEs and microsats?

The Knob180, CentC and TR-1 are the most abundant satellite repeats. It would be interesting to compare these specific families in terms of copy number and arrangement between this assembly and V1.

To what extent are higher order repeats present in these satellite arrays?

The abstract states that centromere length is inversely correlated with CentC satellite repeats. It seems worth noting that CRM retrotransposons are also a major centromere repeat component – and that CENH3 often is not located on CentC?

Line 483-484 – please explain what the definition of the centromere is here? Is this based on CENH3 signal, or on sequence features, or both? Please provide more information on the centromeric Gypsy family (non-CRM) – are these related to ATHILA perhaps, or to other subfamilies?

The diversity of the maize centromeres is very interesting – in many cases there appear to be no satellites in the regions of CENH3 enrichment, whereas in other cases CRM correlates better. In some cases such as Chr10, the CENH3 peak is quite distinct from either CRM or CentC. Is it possible the authors are missing repeat annotation here – eg divergent satellite or CRM, or other repeat families. For example, if this region is analysed using dotplots does this reveal any unannotated repeat arrays? Or is it unique sequence? It would be good to more explicitly compare these patterns to centromeres defined in the Hufford et al assemblies.

Minor points:

In the abstract, what is the definition of 'super long'?

In the abstract, its unclear what 'extreme structural stability' means – do you mean low diversity between 45S copies?

Line 52 – define 'gap' more clearly. Lines 54-55 – the use of gaps here is also unclear. Do you mean gaps in the BAC sequence, or gaps between fully sequenced BACs in the psuedochromosome?

Line 227 – the description of the recombination mechanism here is unclear – please clarify.

Reviewer #2:

Remarks to the Author:

I have read the manuscript by Chen et al. entitled "A complete telomere-to-telomere assembly of maize genome" with great interest. Based on a combination of two long-read sequencing technologies, the authors describe a high-quality assembly of the Mo17 maize genome. They then carried out laborious work to fill in the few remaining gaps and announce the complete sequence of the 10 chromosomes from telomere to telomere. I congratulate the authors for such a strenuous effort to generate this genome assembly.

I have several concerns about, first, the method used to assemble the genome, and finally, the

novelty of the findings presented in the current manuscript. Indeed, I don't think the article will influence thinking in the field, at least not directly in a significant way. As a resource for further work it will likely have an indirect influence. It provides a good example of applying long-read sequencing to non-model organisms, and it is a reminder that most of the genome assemblies that most of us rely on are actually incomplete.

The authors used cutting-edge sequencing technologies, and on the other hand, I think the assembly methods they used are still complicated with many manual steps which are mostly discouraged. Indeed bridging individual reads step by step is not a good approach. And even if they succeed in filling all the gaps, I have serious doubts on the quality of these filled regions. On the contrary, the recently published human T2T genome is based on reliable methods, where corrected HiFi reads are first used to build a contig graph and ultralong nanopore reads are used to untangle the resulting contig graph. In addition, they used long-range data to validate the assembly. Here the authors only used molecules from optical maps, which is not sufficient to validate the structure of their assembly.

Also, using the V1 assembly to organize the ONT long-read contigs is odd, as only 19 out of 567 contigs in the raw assembly were anchored.

In the same way, the validation of the final assembly using ONT reads lead to a figure that exhibits several high- and low-coverage regions (Extended data Figure 15). The authors did not give a robust reason for these non-uniform coverages. I recommend the authors also add the coverage profile of HiFi reads which should show agreement in those particular regions. For example, the 10 high-coverage regions may correspond to collapsed regions which may not have been detected by the kmer completion check.

The combination of the ONT and HiFi assemblies, in particular to fill the gaps ask questions. In gaps 4, 7 and 10, a large part of the ONT contigs did not align to HiFi contigs, and the authors removed those regions in the final assembly, if I understood the process well (first 200Kb of contig6, first 100Kb of contig 13 and first 250Kb of contig 19 in Extended Data Figure 3). Again, the authors should use external data to validate the correct filling of these gaps.

As I said before, the analyzes are mostly descriptive and while they show the impact of generating high-quality assemblies, and represent a valuable resource for the scientific community, I would struggle to identify novel scientific insights described in this manuscript that require evaluation. Centromeres of plants have already been investigated in Naish et al. (2021, Science), Liu et al. (2020, Genome Biology) and Belser et al. (2021, Communications Biology) and tandem repeat arrays in maize genome have already been described in Liu et al. (2020, Genome Biology).

The authors argue that "... telomere to telomere has not been accomplished in a plant genome...", but as an example a banana genome has been assembled recently with several complete chromosomes and remaining gaps were localized in 5S and 45S rDNA clusters and other tandemly repeated sequences (Belser et al. 2021) The number of copies of these repeated elements is often arbitrary due to their length (L235, "... sub-gap2 was determined to be approximately 166 kb"), and therefore leaving a gap or not, does not make a big difference.

Minor points:

- The authors stated that they found sequencing errors previously undocumented in the ONT data, but I think several articles and tools already present this biases :

<https://www.ncbi.nlm.nih.gov/pmc/articles/PMC5600009/>

<https://github.com/natir/yacrd>

<https://www.biorxiv.org/content/10.1101/2022.01.11.475254v1>

- Page 12, line 323 : "The remaining 0.08% of undetected kmers may have been introduced by sequencing errors". This is not informative, as they could also represent unassembled regions.
- Extended Data Figure 3 : "Comparison" instead of Comparison.
- Extended Data Figure 7 : "ONT reads with no sequence errors", do these reads exist and how do you assess that they are error-free ?
- Extended Data Figure 10: "reions" instead of regions
- Extended Data Figure 12: It seems that the Bionano molecules are missing at the bottom of the Figure.
- The figures are of high-quality, but I could not find any information in the methods describing how they were generated and how the TE content of ONT reads was calculated.

Reviewer #3:

Remarks to the Author:

A, B.

Chen et al have produced not just the best assembly to date, but also the most detailed description of structural organization of different diverse repetitive elements of a complex plant genome. This includes ribosomal RNA, telomeres, subtelomeres, heterochromatic knobs, centromeres, MB-scale microsatellite arrays, and segmental gene duplications. They also identify three new satellite sequences and correct one that was identified incorrectly before. Another potentially important contribution is the discovery of potentially systematic problems with ONT reads that had not been described before, including inability to call microsatellite sequences correctly. Chen et al circumvent this problem by using both ONT reads and PacBio HiFi reads.

In summary, this assembly and annotation will be a huge resource for maize genomics and set a new standard for plant genome assembly and annotation going forward.

C.

I have a concern with methodology in that centromere boundaries and centromere size measurements are unreliable because CENH3 ChIP read enrichment was not normalized by control reads. Ideally, this is done by sequencing either the input reads (e.g., Gent et al 2017

<https://pubmed.ncbi.nlm.nih.gov/28637491/>) or IgG control reads (e.g., Altemose et al 2022

<https://pubmed.ncbi.nlm.nih.gov/35357911/>). It can also be done using whole genome sequencing reads as a control (so long as they are processed to the same read length, etc, as in Hufford et al 2021 <https://pubmed.ncbi.nlm.nih.gov/34353948/>). If whole genome reads are not available, even simulated reads can work. Otherwise, CENH3 can be underestimated in highly repetitive regions, such as CentC arrays, and thus peak boundaries called incorrectly.

My second concern is about validation of the quality of gene annotations. Gene annotation is a difficult challenge, particularly with regard to properly identifying features such as translation start sites and transcription start sites. A supplemental figure showing metrics of gene annotation quality compared with other maize genome annotations would be reassuring.

Related to quality validation, I wonder why only ONT reads were used to test for uniformity of read coverage across the final assembly. Since ONT reads were the primary data source of the assembly,

this raises the possibility of circular logic. It would be reassuring to see the HiFi reads used for this purpose.

One other concern relates to data availability: the raw output files from ONT and PacBio can be used to measure methylation across the genome, even in the highly repetitive regions that are invisible to EM-seq or methyl-seq short reads. While methylation analysis is not needed for this work to be published, the raw data, not just the DNA sequence reads, should be made available for others to analyze. Please verify that these are available for both ONT and HiFi data.

D-H. The list below is all minor suggestions and requests for clarifications.

Line 122: 12 Low coverage regions (LCRs) were identified. What about high coverage regions? There appears to be a few in Figure 1B.

Line 152: The chromosome 4 satellite has previously been named Cent4, according to the cited publication (Page et al 2001 <https://pubmed.ncbi.nlm.nih.gov/11560905>)

Line 188: The “three types of previously undocumented sequencing errors” is potentially an important discovery that would be worth expounding on with some more detail. Any estimate of the frequency of the three sorts of errors or what genomic features cause them would be valuable.

Lines 315: What is the difference between fused reads and chimeric reads?

Line 318: What does “threshold of coverage” mean in this context? This term could use a definition.

Line 398: It is surprising that only 15% of segmental gene duplications were tandem duplication. What percent of total segmental duplications were tandem duplications, regardless of genic status?

Line 406: CentC arrays are more likely to be in centromeric regions than pericentromeric. Is there a typo here?

Line 432: Clarify whether sat268 was identified in other 15 NAM founders. Also, there may be an error in logic here: Does presence/absence variation of this array really suggest generation during the divergence of the NAM founders? Or could it also have been segregating in ancestors, and only some of the NAM founders inherited it?

Line 494: CRM is a Gypsy retrotransposon (Sharma and Presting 2014 <https://pubmed.ncbi.nlm.nih.gov/24814286>). On that note, there are four types of CRMs, of which, CRM1 and CRM2 are abundant in centromeres. Which of the four were included in the “CRM” category in this manuscript, and how were they identified?

Line 514: Assuming the same result after normalizing CENH3 enrichment with control reads, does small centromere size suggest CentC provides compactness, or that CentC is a poorer substrate for centromeres? Perhaps CentC invades centromeres in a pathogenic way, not because it is useful.

Line 517: These genes in centromeres may be annotation artefacts, pseudogenes, or gene fragments captured by helitrons. Checking for synteny with sorghum homologs would make them more convincing.

Line 517: Would telomere length be larger in meristem, which does not undergo telomere shortening?

Line 835: What were the 45S and 5S query sequences used for BLAST based quantification of ribosomal DNA abundance?

Line 1083: Were satellite arrays identified using satellite consensus sequences? The citations listed here do not give consensus sequences for CentC, TR-1, or knob180. Consensus sequences for CentC and knob180 were published in Gent et al 2017 <https://pubmed.ncbi.nlm.nih.gov/28637491> and for TR-1 in Liu et al 2020 <https://pubmed.ncbi.nlm.nih.gov/32434565>

Figure 3A: If it could fit in the figure, an inset showing the detailed organization of a few hundred Kb of the knob would be helpful. As it is, the black and red arrowheads are too small to see much on a 20-Mb scale.

Extended Data Figure 1: Which contig was split in two?

Extended Data Figure 2: What is the relation between the red segment and long TAG repeat arrays?

Extended Data Figure 7: Is ONT coverage displayed for the corrected assembly or the basal one? If basal, why not also show coverage for the corrected assembly? Also, the relationship between Extended Data Figure 5 and Extended Data Figure 7 is not clear. One is labeled "Correction" and the other "Validation" but both sets of LCRs appear to have been corrected.

Extended Data Figure 15A would be improved with individual color coded satellites, at least for the abundant ones.

Extended Data Figure 18 Is very confusing. Perhaps the legend is missing important details?

Extended Data Table 5: How is "Homology of NAM founder genes" defined? Related to that, does homology also require syntenic positions in Mo17 relative to NAM founders?

My comments suggest experience in maize genome assembly, tandem repeats, and centromeres, which would naturally point toward a particular maize geneticist as the reviewer. To avoid any mistaken conclusion about reviewer identities, I'll reveal mine here: Jonathan Gent, at your service.

Reviewer #4:

Remarks to the Author:

A complete telomer-to-telomere assembly of the maize genome, Jian Chen et al.

The ms. reports on the resequencing of the maize genome using latest long read sequencing technology and reports on detailed findings/observations of particular structural features that thus far have been largely ignored in traditional genome sequencing/analysis projects. Detailed insights and numbers of rDNA, NORs, telomeric repeats, subtelomeric and subcentromeric repeats as well as the composition of the centromeres are reported. Given the technical difficulties to bridge these in part very long, highly repetitive regions and given the traditional gene-centric view on genomes the work

reported overcomes these limitations and comes up with unprecedented completeness and bridging even the most complex regions. Besides long read technology, exhaustive manual editing has been undertaken and therefore a new quality standard for (plant & crop) genome assemblies is established. Being familiar and involved in plant genomics since >20 years I am familiar with the efforts that have been put into resolving highly repetitive regions. The approach and recipe reported now reports the future blueprint in doing so. While the gene repertoire has been the traditional target in genome analysis this approach/recipe now also allows to target additional structural features!

Criticism:

to be honest it is really hard to find points to target in this ms.!

- I'd be interested in getting some details on the genes hidden in the depth of the centromeres.

Anything special? Any allelic variation? Found in other accessions? Expression? Differentially regulated?

- Figure 1a: Well... The notorious biological decoration. However the reference in the ms. referring to 1a as "assembly of the Mo17 genome" is not really appropriate. Please fix.

- Figs 2a and 2c: please add numbers to the individual bars. Otherwise things look very much alike...

Thanks for the inspiring work

Klaus Mayer

Author Rebuttal to Initial comments

Reviewer #1:

Remarks to the Author:

This paper applies the latest long read sequencing and assembly methods to the maize genome,

creating a 'T2T' map of the Mo17 strain. This does seem like a significant advance but the authors

need to be more explicit about how this assembly is better than Hufford et al 2021 Science, which

reported assemblies of 25 NAM founder inbreds and B73. Also recent B73 Jiao et al 2017 and Mo17

Sun et al 2018 assemblies. I particularly wasn't clear how different the assembly here is to the Mo17

2018 assembly - please spell this out.

**Response:** The T2T Mo17 includes gapless telomere-to-telomere assemblies for all 10

chromosomes of maize, which is an important advance as compared with previous uncomplete

assemblies of maize genomes, including recently reported genome sequences of 25 NAM founder

inbreds and B73 (which all had hundreds of gaps and several unfinished chromosomal ends).

Compared to the reported Mo17ref_V1, this complete assembly added (~85%) or corrected (~15%)

127.15 Mb of sequence that did not linearly align to the pseudomolecules of the Mo17ref_V1. These

were added in the revision. Please see lines 346 to 350, and 363 to 369.

Due to the relevance of maize to plant genetics and agriculture this is an important genome and
makes advances, particularly assembling the first large rDNA clusters in plants (to my knowledge).

The authors generated large (237.7x) ONT and PacBio (69.4x) datasets. Lines 103 to 106 – please
explain the difference between ONT ‘raw ultra long’ and ‘common data’ – what is the difference
between the datasets?

**Response:** The libraries of raw ultra-long ONT data and ONT common data were constructed and
sequenced by different kits of ONT. Compared to the ultra-long data with read N50 of 73.7 kb, the
read N50 of common data was only 28.6 kb. This was added in the revision to make it clear.

I note the authors used Mo17ref_V1 to ‘anchor and orientate’ with – have the authors tried reference
free methods of orientating/scaffolding? Could any possible errors in the V1 assembly be carried
over at this step?

**Response:** In consideration of long length of the contig of ONT based Mo17 assembly (N50 = 181.2
33 Mb), individual possible anchor and orient errors of contigs (N50 = 1.48 Mb) of the Mo17ref_V1
should not affect pseudomolecules construction here. In addition, the anchor and orient of finally
used 20 contigs for pseudomolecules construction were verified by GBS tags, which were added in
the revision (lines 118 to 120). The successful closure of all gaps also reflected the validity of anchor
and orient of the pseudomolecules.

Lines 120-123 – please define ‘fairly uniform coverage’ with respect to ONT read alignments to the
assembly. Did the authors also try aligning PacBio HiFi reads to the assembly in a similar way?

Please succinctly explain what the low coverage regions are in more detail (lines 122-123).

**Response:** According to alignments of ultra-long ONT reads > 10 kb, there were only 12 low
coverage regions (LCRs) with read depth lower than 100 and 10 high coverage regions (HCRs) with
read depth higher than 250. The coverage of remained genomic regions were relatively uniform,
with an average of 177x. These, as well as the detail of LCRs and HCRs were added in the revision
(lines 126 to 130, and 157 to 173). Similar coverage analyses were performed for the final assembly
with PacBio HiFi reads (Please see lines 321 to 325).

Lines 129-130 – please provide more detail on how the PacBio contigs and ONT assembly were
integrated – how congruent were the individual assemblies before integration? In the case of
differences, how do you decide which alternative is correct?

**Response:** The contigs of both PacBio Hifiasm and Canu assemblies were aligned onto the basal
Mo17 assembly. Only the PacBio contigs which can span the gaps or correct the assembly errors of
the basal Mo17 assembly were integrated with the ONT-based assembly. The assemblies of all these
regions closed or corrected by the PacBio contigs were confirmed by the uniform ONT-read
coverage and tiling of ONT reads (lines 145 to 147).

Several recent papers, for example in human and Arabidopsis, have reported successful gapless
assembly of centromere repeat arrays. However, the 5S and 45S rDNA have remained very
challenging. The authors propose to have successfully assembled these arrays in maize. To validate
these assemblies, a ‘blast-based copy number estimation’ is performed – please provide more
explanation – I'm unclear how this provides validation? I'm also unclear why the authors think that
k-mer based estimates from Illumina data would likely underestimate rDNA copy number? I can
appreciate that there would be many identical k-mers due to the low variation, but would this not be
reflected in a higher copy number? In any case, as the authors have Illumina reads for this strain, it
would be interesting to see how estimates using this method compared to the copy number estimates
from the assembly. Line 455 – which TEs were found within the 5S – were they different from the
surrounding regions?

**Response:** For the blast-based copy number estimation, the sequences of rDNAs were aligned to
the ONT ultra-long and PacBio HiFi data to identified all rDNA sequences harbored in the data,
which were then then normalized by the length of rDNA repeat unit and the average genome
coverage of the data to estimate the copy number of rDNA in the genome. Please see lines 876 to
884 in the method section. We applied a method (lines 885 to 895) to estimate the copy number of
5S and 45S rDNAs using the Illumina data in the revision. Corresponding results (lines 151 to 153,
and 301 to 303) were added in the revision. Compared to the flanking regions, TEs inserted in the
5S rDNA array were enriched with Copia elements (lines 516 to 518).

The rDNA array structures shown in Fig 4D and 4E are very interesting and provide a potential first

look inside complete plant rDNA clusters - the 'gradient' of TEs and rDNA variation seen the 5S
cluster is particularly interesting.

One major suggestion I have for the paper, is that as the authors have abundant ONT data, and
Deepsignal-plant has been proven to work well for plant repeat regions (eg Naish et al 2021), it
would be beneficial to the study if DNA methylation maps across the T2T genome could be
presented in this study. For example, plotting methylation on the very interesting rDNA assemblies
in Fig 4D and 4E would be very interesting. The paper also reveals large (up to 235 kp) tri-nucleotide
repeats that are validated cytogenetically. Equally, I would be interested to know the methylation
state of these repeats.

**Response:** Thank you for your suggestions. Indeed, the raw output files of the ONT are useful for
measuring DNA methylation. As you mentioned, it's interesting to get some features of DNA
methylation across the rDNA arrays and large trinucleotide repeats arrays, which the methylation
features had not been glanced before. Besides, it's also meaningful to survey methylation profiles
across other types of highly repetitive regions, including TR-1, Knobs, centromeres, subtelomeres,
as well as some TE regions which methylation could not be well explored using typical MethylC-
seq data. According to your suggestion, we plan to perform a comprehensive analysis of methylation
features of the T2T genome and present the results in the future.

Line 131 – which transposon families are involved at these regions?

**Response:** It is hard to define the exact regions associated with these gaps. Based on analysis of
300 kb flanking regions of the three TE-related gaps, no particular transposon families were found
to be involved at these regions.

Line 153 – how do the authors know that ONT is introducing 'extra' sequence errors in TAG repeats?

Line 157 – I'm not clear how you know which reads are mistakenly read, and which are not?

**Response:** As reported previously (Tan et al, Biorxiv, 2022), here we found a stretch of
microsatellites in ONT reads could be miscalled as other microsatellites, including many reads with
long TAG repeats or telomeric repeats. Compared to the 'right called' reads, the 'miscalled' reads
showed significantly lower base quality at the regions with TAG repeats (Extended Data Figure 8).

An iterative alignment method using PacBio data is reported to assemble the 45S, ultimately

building 2,974 copies. I'm unclear how the authors can be confident in this assembly although the
read tiling path shown looks convincing.

**Response:** Here, we provide the first completely assembled genomic structure of such a tens of
megabases-scale 45S rDNA array (26.8 Mb) in maize. Indeed, we did not find an independent
method to evaluate the assembly of entire region of the 45S array. However, as mentioned in the
manuscript, some aspects of the array were evaluated, including the number of 45S rDNAs and the
assembly of a 1.2 Mb TE-enriched region, which reflected the right assembly of 45S rDNA array.

Lines 304 – 307 – what is the cause of the high read coverage mapping here? This would seem to
indicate missing sequences/copies from the assembly, causing a stack up of reads on the copies that
are present. With respect to the mitochondrial insertions – recent work suggests that nuclear mtDNA
sequences are likely to be DNA methylated, whereas the organelle copies will not be. Therefore,
could the ONT reads be assessed for methyl state, as a means to restrict assembly to those most
likely to represent nuclear insertions (assuming they are methylated).

See <https://www.biorxiv.org/content/10.1101/2022.02.22.481460v1>

**Response:** There were 10 HCRs (identified using ONT reads > 10 kb, totally 177×), 5 of which
were related to genomic regions with homolog sequences at the mitochondrial genome, 4 of which
were related to TEs, and one was related to subtelomeric repeat array longer than 150 kb. With the
ONT reads longer than 50 kb (totally 123×), the coverage of 9 HCRs was relatively uniform, except
for one HCR (HCR 8) with 2 copies of super-long tandem repeat unit (about 300 kb) (lines 163 to
173). For HCR 8, an obviously complementary coverage distribution was observed between the 2
copies of super-long tandem repeat unit for ONT reads longer than 50 kb (Supplementary data figure
9). These results suggested that high coverage of HCRs was introduced by mapping errors of reads
from their corresponding homolog sequences. In addition, the assemblies of all HCRs were
confirmed by concordant PacBio assembly and tiling ONT reads (lines 157 to 158).

Line 158 – I'm surprised that this inversion has not been observed previously? For example, has
crossover suppression been observed within this inversion in genetic maps made between Mo17
and strains that lack the inversion? Could it be possible that the inversion is polymorphic between
Mo17 strains used for different genome assemblies?

**Response:** The basal Mo17 assembly in the inversion was validated by the concordant PacBio

assembly and uniform ONT-read coverage, and was consistent with the assemblies all 25 NAM
founder lines and B73 inbred. This suggested that the inversion was caused by possible earlier
anchoring and orientation errors for contigs of the Mo17ref_V1 assembly. The Mo17ref_V1 was
anchored and oriented using GBS tags. The density of available GBS tags was obviously low in the
inversion (Extended Data Figure 1).

For the telomeres, how do these size estimates compare to published work on the maize telomeres?
Certainly in many plants telomere size has been estimated by Southern blotting (eg work of Shippen
lab) – perhaps this data exists for maize and can be compared to?

**Response:** Thank you for your suggestion. The average length of 20 telomeres of the complete T2T
Mo17 genome was 26.1 kb, longer than average telomere size (11.7 kb, ranging from 1.8 to 40.0
152 kb) of 22 maize inbreds estimated by Southern blotting (Burr, et al., Plant Cell, 1992). The
153 comparison was added in the revision.

Lines 338 – 342 – how many of these sequences are not represented in the previous Mo17 assembly?
Were some expanded more than others in the new assembly relative to V1? Where do the new
regions locate on the chromosomes? How many are centromeric? Line 344 – define ‘colocalized’
here more clearly – what relationship are the authors proposing between the TEs and microsats?

**Response:** Compared to the Mo17ref_V113, about 3.4 Mb microsattellites (73.4% were in 5 super-
long TAG repeat arrays), 33.3 Mb satellites (68.4% were in two knobs at chr6S and 8L, and 19.6%
were in centromeric regions), and almost all (99.8%) rDNAs were newly assembled here. This was
added in the revision (lines 363 to 366). ‘Colocalized’ means that a part of microsattellites,
minisattellites, and satellites were identified in TE regions. We rewrote this sentence to make it clear.
Please see line 362.

The Knob180, CentC and TR-1 are the most abundant satellite repeats. It would be interesting to
compare these specific families in terms of copy number and arrangement between this assembly
and V1.

**Response:** As only 5.60% of knob180, 14.74% of TR-1, 4.13% of CentC repeats were assembled
in the Mo17ref_V1 (lines 427 to 428), we did not compare the arrangement between the T2T Mo17
assembly and Mo17ref_V1.

To what extent are higher order repeats present in these satellite arrays?

**Response:** Some higher order repeats analyses were added in the revision. Please see lines 449 to
464.

The abstract states that centromere length is inversely correlated with CentC satellite repeats. It
seems worth noting that CRM retrotransposons are also a major centromere repeat component – and
that CENH3 often is not located on CentC?

**Response:** In previous version of the manuscript, the published CENH3 ChIP-seq data, which
corresponding control data (input reads) is unavailable, was used to calculate CENH3 level and
define centromeres. To obtain the reliable CENH3 enrichment level and avoid the possible
polymorphic difference between the Mo17 strains, we regenerated the ChIP-seq data of CENH3
and input data here, and redefined the centromeres. Weak inverse correlation ($R^2 = 0.19$, $P = 0.205$)
was observed between centromere and CentC lengths. So corresponding result was removed in the
revision.

Line 483-484 – please explain what the definition of the centromere is here? Is this based on CENH3
signal, or on sequence features, or both? Please provide more information on the centromeric Gypsy
family (non-CRM) – are these related to ATHILA perhaps, or to other subfamilies?

**Response:** Here we defined the centromeric regions of the T2T Mo17 genome based on the
enrichment level of CENH3. The centromeric Gypsy elements (non-CRM) are mainly consisted by
Ciful-Zeon, Flip, Prem1 families (line 539).

The diversity of the maize centromeres is very interesting – in many cases there appear to be no
satellites in the regions of CENH3 enrichment, whereas in other cases CRM correlates better. In
some cases such as Chr10, the CENH3 peak is quite distinct from either CRM or CentC. Is it
possible the authors are missing repeat annotation here – eg divergent satellite or CRM, or other
repeat families. For example, if this region is analysed using dotplots does this reveal any unannotated
repeat arrays? Or is it unique sequence? It would be good to more explicitly compare these patterns
to centromeres defined in the Hufford et al assemblies.

**Response:** All sequences around the centromeres were annotated using RepeatMasker, and no

repeats were missed or unmarked in Figure 5. Thank you for your suggestion, the patterns of
centromeric sequences compositions were discussed and compared with that of NAM founders
(Wang et al. Genetics, 2021). Please see lines 550 to 558.

Minor points:

In the abstract, what is the definition of ‘super long’?

**Response:** Here tri-nucleotide repeat arrays longer than 200 kb was termed as ‘super-long’. We
rewrote the sentence to make it clear.

In the abstract, its unclear what ‘extreme structural stability’ means – do you mean low diversity
between 45S copies?

**Response:** Extreme structural stability means exceptionally low level of local inversion and TE
insertion in the 26.8 Mb nucleolar organizer region. We rewrote the sentence to make it clear.

Line 52 – define ‘gap’ more clearly. Lines 54-55 – the use of gaps here is also unclear. Do you mean
gaps in the BAC sequence, or gaps between fully sequenced BACs in the psuedochromosome?

**Response:** We rewrote these sentences to make it clear. Please see lines 53.

Line 227 – the description of the recombination mechanism here is unclear – please clarify.

**Response:** We rewrote these sentences to make it clear. Please see lines 245 to 246.

Reviewer #2:

Remarks to the Author:

I have read the manuscript by Chen et al. entitled “A complete telomere-to-telomere assembly of
maize genome” with great interest. Based on a combination of two long-read sequencing
technologies, the authors describe a high-quality assembly of the Mo17 maize genome. They then
carried out laborious work to fill in the few remaining gaps and announce the complete sequence of
the 10 chromosomes from telomere to telomere. I congratulate the authors for such a strenuous
effort to generate this genome assembly.

I have several concerns about, first, the method used to assemble the genome, and finally, the

novelty of the findings presented in the current manuscript. Indeed, I don't think the article will
influence thinking in the field, at least not directly in a significant way. As a resource for further
work it will likely have an indirect influence. It provides a good example of applying long-read
sequencing to non-model organisms, and it is a reminder that most of the genome assemblies that
most of us rely on are actually incomplete.

**Response:** Indeed, the complete telomere-to-telomere assembly of the Mo17 genome is the most
remarkable achievement for our study. Based on the assembly, we also revealed some previously
uncharacterized genome features. For example, the very first fully assembled 45S rDNA array (a
total length of 26.8 Mb, with 2,974 45S rDNA copies) displayed with exceptionally low level of
local inversion and TE insertion. There are some super-long TAG repeat arrays (> 200 kb) in the
genome. Certainly, the functional role of these features remains to be determined.

The authors used cutting-edge sequencing technologies, and on the other hand, I think the assembly
methods they used are still complicated with many manual steps which are mostly discouraged.
Indeed bridging individual reads step by step is not a good approach. And even if they succeed in
filling all the gaps, I have serious doubts on the quality of these filled regions. On the contrary, the
recently published human T2T genome is based on reliable methods, where corrected HiFi reads
are first used to build a contig graph and ultralong nanopore reads are used to untangle the resulting
contig graph. In addition, they used long-range data to validate the assembly. Here the authors only
used molecules from optical maps, which is not sufficient to validate the structure of their assembly.

**Response:** Here we first obtained the basal Mo17 assembly by NextDenovo assembler using ultra-
long ONT data. Then, five gaps and six LCRs with assembly errors in the ONT based assembly
were closed or corrected by PacBio HiFi data based assemblies obtained with Canu and Hifiasm
assemblers. Only 5 gaps related to super-long TAG repeat arrays (average 784.3 kb, about 66%
sequences are TAG repeats) and 1 gap related to a tens of megabases-scale 45S rDNA array (26.8
259 Mb), which were failed to be assembled by automatic assembler, were closed by manual. In addition,
the assembly quality of these six manually closed gaps were all evaluated. Duo to the greatly
differences between human and maize genomes, the methods used for assembly of human genome
were unsuitable for individual regions of maize genome, such as the tens of megabases-scale 45S
rDNA array.

Also, using the V1 assembly to organize the ONT long-read contigs is odd, as only 19 out of 567
contigs in the raw assembly were anchored.

**Response:** Totally, 428 ONT contigs were anchored according to the alignment with the backbone
of Mo17ref_V1. Among them, 19 contigs, which were non-redundant with each other and could
cover all aligned genomic regions, were selected to construct pseudomolecules contigs. This were
added in the revision to make it clear (lines 115 to 117). In fact, for all 567 ONT contigs, 19 contigs
were used for constructing pseudomolecules, 533 contigs were redundancies for the constructed
pseudomolecules, 14 were misassembled contigs, and 1 was introduced duo to the contamination
of bacteria.

In the same way, the validation of the final assembly using ONT reads lead to a figure that exhibits
several high- and low-coverage regions (Extended data Figure 15). The authors did not give a robust
reason for these non-uniform coverages. I recommend the authors also add the coverage profile of
HiFi reads which should show agreement in those particular regions. For example, the 10 high-
coverage regions may correspond to collapsed regions which may not have been detected by the
kmer completion check.

**Response:** All local coverage-anomalous regions in the finally assembly (Extended data Figure 18)
were corresponded to gaps, LCRs and HCRs identified based on the basal Mo17 assembly, which
the cause of these non-uniform coverages was indicated in the revision (lines 133 to 173). The
coverage profile of HiFi reads in these regions were also added in the revision (Supplementary data
figure 5, 6, 7, and 9.

The combination of the ONT and HiFi assemblies, in particular to fill the gaps ask questions. In
gaps 4, 7 and 10, a large part of the ONT contigs did not align to HiFi contigs, and the authors
removed those regions in the final assembly, if I understood the process well (first 200Kb of contig6,
first 100Kb of contig 13 and first 250Kb of contig 19 in Extended Data Figure 3). Again, the authors
should use external data to validate the correct filling of these gaps.

**Response:** Only the PacBio contigs which can span the gaps or correct the assembly errors of the
basal Mo17 assembly were integrated with the ONT-based assembly. The assemblies of all these
regions closed or corrected by the PacBio contigs, including gaps 4, 7 and 10, were confirmed by
the uniform ONT-read coverage and tiling of ONT reads. Please see lines 145 to 147.

As I said before, the analyzes are mostly descriptive and while they show the impact of generating
high-quality assemblies, and represent a valuable resource for the scientific community, I would
struggle to identify novel scientific insights described in this manuscript that require evaluation.
Centromeres of plants have already been investigated in Naish et al. (2021, Science), Liu et al. (2020,
Genome Biology) and Belser et al. (2021, Communications Biology) and tandem repeat arrays in
maize genome have already been described in Liu et al. (2020, Genome Biology).

**Response:** Indeed, some characteristics of centromeres and tandem repeat arrays have been
described by Liu et al. (2020, Genome Biology). However, due to fail of the assembly of some
satellite arrays (including Knob-6S and Knob-8L) and several centromeres (including chromosomes
1, 6, 7), some related features had not been revealed. Here we firstly presented a structural view of
all centromeres and tandem repeat arrays in the maize genome, including sequence compositions of
all ten centromeres, and Knob-6S and Knob-8L (the two knobs taken up the majority knob180 and
TR-1 repeats in the genome). The studies of Naish et al. (2021, Science) and Belser et al. (2021,
Communications Biology) focused on the genomes of Arabidopsis and banana, respectively, which
the features of centromeres are largely different with that of maize. For example, compared to
Arabidopsis centromeres mainly composed by CEN180, maize centromeres are mainly consisted
by CentC, CRM, and Gypsy retrotransposons. Abundant 5S rDNAs were identified in the
centromeres of banana, but there was no any 5S rDNA in maize centromeres. Besides, we also
revealed some other previously uncharacterized genome features. For example, we found that the
very first fully assembled 45S rDNA array (a total length of 26.8 Mb, with 2,974 45S rDNA copies)
displayed with exceptionally low level of local inversion and TE insertion, and that there are some
super-long TAG repeat arrays (> 200 kb) in the maize genome.

The authors argue that "... telomere to telomere has not been accomplished in a plant genome...",
but as an example a banana genome has been assembled recently with several complete
chromosomes and remaining gaps were localized in 5S and 45S rDNA clusters and other tandemly
repeated sequences (Belser et al. 2021) The number of copies of these repeated elements is often
arbitrary due to their length (L235, "... sub-gap2 was determined to be approximately 166 kb"), and
therefore leaving a gap or not, does not make a big difference.

**Response:** This inappropriate description has been deleted in the revision.

Minor points:

- The authors stated that they found sequencing errors previously undocumented in the ONT data,
but I think several articles and tools already present this biases :

<https://www.ncbi.nlm.nih.gov/pmc/articles/PMC5600009/>

<https://github.com/natir/yacrd>

<https://www.biorxiv.org/content/10.1101/2022.01.11.475254v1>

**Response:** Thank you for your reminding. Corresponding articles were referred in the revision.

- Page 12, line 323: “The remaining 0.08% of undetected kmers may have been introduced by
sequencing errors”. This is not informative, as they could also represent unassembled regions.

**Response:** For the remaining 0.08% of undetected k-mers, about 60% of which were introduced by
base errors within reads or assembly, 30% of which were introduced by accumulation of multiple
similar base errors of reads from several different genomic regions, and a small amount of which
were possible exogenous DNA contamination failed to be excluded. This were added in the revision.

- Extended Data Figure 3 : “Comparation” instead of Comparison.

**Response:** It’s done.

- Extended Data Figure 7: “ONT reads with no sequence errors”, do these reads exist and how do
you assess that they are error-free ?

**Response:** As reported previously (Tan et al, Biorxiv, 2022), here we found a stretch of
microsatellites in ONT reads could be miscalled as other microsatellites, including many reads with
long TAG repeats or telomeric repeats (lines 210 to 212). Compared to the ‘right called’ reads, the
‘miscalled’ reads showed significantly lower base quality at the regions with TAG repeats
(Extended Data Figure 8). We have revised this sentence to make it clear.

- Extended Data Figure 10: “reions” instead of regions

**Response:** It’s done.

- Extended Data Figure 12: It seems that the Bionano molecules are missing at the bottom of the
Figure.

**Response:** We are sorry for the negligence. It was corrected in the revision.

- The figures are of high-quality, but I could not find any information in the methods describing
how they were generated and how the TE content of ONT reads was calculated.

**Response:** Thank you. The sequence composition for given genomic regions or ONT reads were
annotated using RepeatMasker. Then, different colors were assigned to different types of repeats,
including different TEs, for graphical represent. This was added in the revision. Please see lines
1037 to 1047.

Reviewer #3:

Remarks to the Author:

371 A, B.

Chen et al have produced not just the best assembly to date, but also the most detailed description
of structural organization of different diverse repetitive elements of a complex plant genome. This
includes ribosomal RNA, telomeres, subtelomeres, heterochromatic knobs, centromeres, MB-scale
microsatellite arrays, and segmental gene duplications. They also identify three new satellite
sequences and correct one that was identified incorrectly before. Another potentially important
contribution is the discovery of potentially systematic problems with ONT reads that had not been
described before, including inability to call microsatellite sequences correctly. Chen et al
circumvent this problem by using both ONT reads and PacBio HiFi reads.

In summary, this assembly and annotation will be a huge resource for maize genomics and set a
new standard for plant genome assembly and annotation going forward.

C.

I have a concern with methodology in that centromere boundaries and centromere size
measurements are unreliable because CENH3 ChIP read enrichment was not normalized by control
reads. Ideally, this is done by sequencing either the input reads (e.g., Gent et al
2017 <https://pubmed.ncbi.nlm.nih.gov/28637491>) or IgG control reads (e.g., Altemose et al
2022 <https://pubmed.ncbi.nlm.nih.gov/35357911>). It can also be done using whole genome

sequencing reads as a control (so long as they are processed to the same read length, etc, as in
Hufford et al 2021 <https://pubmed.ncbi.nlm.nih.gov/34353948>). If whole genome reads are not
available, even simulated reads can work. Otherwise, CENH3 can be underestimated in highly
repetitive regions, such as CentC arrays, and thus peak boundaries called incorrectly.

**Response:** Thank you for your reminding. We had regenerated the ChIP-seq data of CENH3 and
corresponding control data in the revision. The centromere boundaries were redetermined based on
the enrichment level of CENH3 (normalized by control data).

My second concern is about validation of the quality of gene annotations. Gene annotation is a
difficult challenge, particularly with regard to properly identifying features such as translation start
sites and transcription start sites. A supplemental figure showing metrics of gene annotation quality
compared with other maize genome annotations would be reassuring.

**Response:** The metrics of gene annotated here were compared with genes annotated in the
Mo17ref_v1, B73ref_v4, and B73ref_v5 in the revision (Extended Data Table 5). The quality of
gene annotations is comparable with that of B73 gene annotations.

Related to quality validation, I wonder why only ONT reads were used to test for uniformity of read
coverage across the final assembly. Since ONT reads were the primary data source of the assembly,
this raises the possibility of circular logic. It would be reassuring to see the HiFi reads used for this
purpose.

**Response:** PacBio HiFi reads were also used for quality validation in the revision. Please see lines
321 to 325.

One other concern relates to data availability: the raw output files from ONT and PacBio can be
used to measure methylation across the genome, even in the highly repetitive regions that are
invisible to EM-seq or methyl-seq short reads. While methylation analysis is not needed for this
work to be published, the raw data, not just the DNA sequence reads, should be made available for
others to analyze. Please verify that these are available for both ONT and HiFi data.

**Response:** The raw output bam files of PacBio and fast5 files of ONT are indeed helpful for
measuring DNA methylation, particularly in highly repetitive regions. Regrettably, we did not copy
the raw output bam files of PacBio from the sequencing company when the data were generated (at

2020), and these raw files were deleted by the company now. The raw output fast5 files of ONT
data generated here are available. There was a total of about 19 Tb raw fast5 files for ONT data,
and it is difficult to upload such a large data to public database. Certainly, all raw output files of
ONT data will be available when required.

D-H. The list below is all minor suggestions and requests for clarifications.

Line 122: 12 Low coverage regions (LCRs) were identified. What about high coverage regions?
There appears to be a few in Figure 1B.

**Response:** Analysis of high coverage regions was added in the revision. Please see lines 167 to 173.

Line 152: The chromosome 4 satellite has previously been named Cent4, according to the cited
publication (Page et al 2001 <https://pubmed.ncbi.nlm.nih.gov/11560905>)

**Response:** Thank you for your reminding. 'Cent4' was used in the revision.

Line 188: The “three types of previously undocumented sequencing errors” is potentially an
important discovery that would be worth expounding on with some more detail. Any estimate of
the frequency of the three sorts of errors or what genomic features cause them would be valuable.

**Response:** Based on the ultra-long ONT reads longer than 10 kb, about 3.6% of which were fused
reads, and 1.5% were symmetrical reads (lines 331 to 332). Regrettably, we failed to find a method
to calculate the proportion of microsatellites miscalled reads.

Lines 315: What is the difference between fused reads and chimeric reads?

**Response:** Here chimeric reads refer the reads of unknown mistaken origin, which were not grouped
as fused reads, symmetrical reads, microbial reads, as well as reads originated from maize
mitochondrion and chloroplast genomes and microbial DNA contamination based on the thresholds
we used. We rewrote this sentence to make it clear (line 332).

Line 318: What does “threshold of coverage” mean in this context? This term could use a definition.

**Response:** We rewrote this sentence to make it clear (line 334).

Line 398: It is surprising that only 15% of segmental gene duplications were tandem duplication.

What percent of total segmental duplications were tandem duplications, regardless of genic status?

**Response:** For segmental duplications without genes, the proportion of tandem segmental
duplication is about 12%. This was added in the revision.

Line 406: CentC arrays are more likely to be in centromeric regions than pericentromeric. Is there
a typo here?

**Response:** CentC arrays were all located in centromeric and pericentromeric regions. It has been
corrected in the revision.

Line 432: Clarify whether sat268 was identified in other 15 NAM founders. Also, there may be an
error in logic here: Does presence/absence variation of this array really suggest generation during
the divergence of the NAM founders? Or could it also have been segregating in ancestors, and only
some of the NAM founders inherited it?

**Response:** This presence/absence variation might have been generated in their ancestors as only
three of 19 wild relatives were found with sat 268 repeats according to resequencing data from
maize Hapmap2. This was added in the revision.

Line 494: CRM is a Gypsy retrotransposon (Sharma and Presting
2014 <https://pubmed.ncbi.nlm.nih.gov/24814286>). On that note, there are four types of CRMs, of
which, CRM1 and CRM2 are abundant in centromeres. Which of the four were included in the
"CRM" category in this manuscript, and how were they identified?

**Response:** The four types of CRMs, including CRM1, CRM2, CRM3 and CRM4, were all used for
identification of CRM using RepeatMasker. Please see lines 1020 to 1022 in the Method section.

Line 514: Assuming the same result after normalizing CENH3 enrichment with control reads, does
small centromere size suggest CentC provides compactness, or that CentC is a poorer substrate for
centromeres? Perhaps CentC invades centromeres in a pathogenic way, not because it is useful.

**Response:** In previous version of the manuscript, the published CENH3 ChIP-seq data, which
corresponding control data (input reads) is unavailable, was used to calculate CENH3 level and
define centromeres. To obtain the reliable CENH3 enrichment level and avoid the possible
polymorphic difference between the Mo17 strains, we regenerated the ChIP-seq data of CENH3

and input data here, and redefined the centromeres. Weak inverse correlation ($R^2 = 0.19$, $P = 0.205$)
was observed between centromere and CentC lengths. So corresponding result was removed in the
revision.

Line 517: These genes in centromeres may be annotation artefacts, pseudogenes, or gene fragments
captured by helitrons. Checking for synteny with sorghum homologs would make them more
convincing.

**Response:** Totally, 82 genes were identified in centromeres of the Mo17 genome, which most (72)
had homolog genes in maize NAM founder lines and 52 had homolog genes in sorghum. These
were added in the revision.

Line 517: Would telomere length be larger in meristem, which does not undergo telomere
shortening?

**Response:** A quantitative analysis of telomere length of single cells in Arabidopsis root apex
uncovered a heterogeneous telomere-length distribution of different cell lineages, with the longest
telomeres at the stem cells (Mary-Paz González-García, et al., Cell Reports, 2015). Proper
telomere maintenance in stem cells is essential for their ability to sustain meristem growth. However,
the correlation between telomere length and meristem activity remains to be determined in maize.

Line 835: What were the 45S and 5S query sequences used for BLAST based quantification of
ribosomal DNA abundance?

**Response:** The rDNA query sequences used for BLAST based copy number evaluation were added
in the revision. Please see lines 879 to 882.

Line 1083: Were satellite arrays identified using satellite consensus sequences? The citations listed
here do not give consensus sequences for CentC, TR-1, or knob180. Consensus sequences for CentC
and knob180 were published in Gent et al 2017 <https://pubmed.ncbi.nlm.nih.gov/28637491> and for
TR-1 in Liu et al 2020 <https://pubmed.ncbi.nlm.nih.gov/32434565>

**Response:** The consensus sequences of five previously reported satellites (knob180, TR-1, CentC,
Cent4, and tRNASAT_ZM) used for blast were added in the revision. Please see lines 1175 to 1179.

Figure 3A: If it could fit in the figure, an inset showing the detailed organization of a few hundred
Kb of the knob would be helpful. As it is, the black and red arrowheads are too small to see much
on a 20-Mb scale.

**Response:** Thank you for your suggestion. It's done. Pleased see the revised Figure 3A.

Extended Data Figure 1: Which contig was split in two?

**Response:** The contig 3 and contig 6 showed in Extended Data Figure 1 were generated by split of
a raw ONT contig with assembly error (see Extended Data Figure 2). This was added in the legend
of Extended Data Figure 1.

Extended Data Figure 2: What is the relation between the red segment and long TAG repeat arrays?

**Response:** The red box represents the misassembled region related to two TAG array related gaps
on chromosome 1 (Gap 2) and 2 (Gap 5). This was added in the legend of Extended Data Figure 2.

Extended Data Figure 7: Is ONT coverage displayed for the corrected assembly or the basal one? If
basal, why not also show coverage for the corrected assembly? Also, the relationship between
Extended Data Figure 5 and Extended Data Figure 7 is not clear. One is labeled "Correction" and
the other "Validation" but both sets of LCRs appear to have been corrected.

**Response:** The coverage of both the basal and corrected assemblies was showed in the revised
Extended Data Figure 7. 'Correction' is now used in the legend of Extended Data Figure 7 to make
it clear.

Extended Data Figure 15A would be improved with individual color coded satellites, at least for the
abundant ones.

**Response:** CentC, TR1, knob180, and other satellite were individually colored in the in the revised
figure (Extended Data Figure 17 in the revision).

Extended Data Figure 18 Is very confusing. Perhaps the legend is missing important details?

**Response:** We rewrote the legend of this figure (Extended Data Figure 19 in the revision) to make
it clear.

Extended Data Table 5: How is “Homology of NAM founder genes” defined? Related to that, does
homology also require syntenic positions in Mo17 relative to NAM founders?

**Response:** Homology analysis was performed with OrthoFinder based on the protein sequences of
genes annotated in Mo17 and NAM founder lines. The genes that can be classified into Orthogroups
with least one gene of NAM founder lines were defined as homologies of NAM founder genes.
Syntenic positions are not required. These were added in the section of ‘Methods’.

My comments suggest experience in maize genome assembly, tandem repeats, and centromeres,
which would naturally point toward a particular maize geneticist as the reviewer. To avoid any
mistaken conclusion about reviewer identities, I’ll reveal mine here: Jonathan Gent, at your service.

Reviewer #4:

Remarks to the Author:

A complete telomer-to-telomere assembly of the maize genome, Jian Chen et al.

The ms. reports on the resequencing of the maize genome using latest long read sequencing
technology and reports on detailed findings/observations of particular structural features that thus
far have been largely ignored in traditional genome sequencing/analysis projects. Detailed insights
and numbers of rDNA, NORs, telomeric repeats, subtelomeric and subcentromeric repeats as well
as the composition of the centromeres are reported. Given the technical difficulties to bridge these
in part very long, highly repetitive regions and given the traditional gene-centric view on genomes
the work reported overcomes these limitations and comes up with unprecedented completeness and
bridging even the most complex regions. Besides long read technology, exhaustive manual editing
has been undertaken and therefore a new quality standard for (plant & crop) genome assemblies is
established. Being familiar and involved in plant genomics since >20 years I am familiar with the
efforts that have been put into resolving highly repetitive regions. The approach and recipe reported
now reports the future blueprint in doing so. While the gene repertoire has been the traditional target
in genome analysis this approach/recipe now also allows to target additional structural features!

**Response:** Thank you for your encouraging comments.

Criticism:

to be honest it is really hard to find points to target in this ms.!

- I'd be interested in getting some details on the genes hidden in the depth of the centromeres.

Anything special? Any allelic variation? Found in other accessions? Expression? Differentially

regulated?

**Response:** A total of 82 genes were identified in centromeres of the Mo17 genome, which most (72)

had homolog genes in maize NAM founder lines and 52 had homolog genes in sorghum. The

centromeric genes relatively preferred to be tissue specifically expressed. These were added in the

revision.

- Figure 1a: Well... The notorious biological decoration. However the reference in the ms. referring to

1a as "assembly of the Mo17 genome" is not really appropriate. Please fix.

**Response:** The sentence was rewritten to make it appropriate for referring figure 1a. Please see lines

104 to 105.

- Figs 2a and 2c: please add numbers to the individual bars. Otherwise things look very much alike...

**Response:** Revised accordingly.

Thanks for the inspiring work

Klaus Mayer

Decision Letter, first revision:

12th Oct 2022

Dear Professor Lai,

Your Article, "A complete telomere-to-telomere assembly of the maize genome" has now been seen by 4 referees. You will see from their comments below that while they find your work of interest, some important points are raised. We are interested in the possibility of publishing your study in Nature Genetics, but would like to consider your response to these concerns in the form of a revised manuscript before we make a final decision on publication.

To guide the scope of the revisions, the editors discuss the referee reports in detail within the team with a view to identifying key priorities that should be addressed in revision. In this case, we think all

the referees have identified important aspects of the analyses that need to be addressed or improved. In addition, we strongly encourage you to release the source fast5 files of ONT data in a public database (Reviewer #3). We ask that you address all referee comments as thoroughly as possible with appropriate revisions.

We therefore invite you to revise your manuscript taking into account all reviewer and editor comments. Please highlight all changes in the manuscript text file. At this stage we will need you to upload a copy of the manuscript in MS Word .docx or similar editable format.

*2) If you have not done so already please begin to revise your manuscript so that it conforms to our Article format instructions, available [here](http://www.nature.com/ng/authors/article_types/index.html). Refer also to any guidelines provided in this letter.

[redacted]

We hope to receive your revised manuscript within 3 to 6 months. If you cannot send it within this time, please let us know.

Nature Genetics is committed to improving transparency in authorship. As part of our efforts in this

direction, we are now requesting that all authors identified as 'corresponding author' on published papers create and link their Open Researcher and Contributor Identifier (ORCID) with their account on the Manuscript Tracking System (MTS), prior to acceptance. ORCID helps the scientific community achieve unambiguous attribution of all scholarly contributions. You can create and link your ORCID from the home page of the MTS by clicking on 'Modify my Springer Nature account'. For more information please visit www.springernature.com/orcid.

Sincerely,

Wei Li, PhD
Senior Editor
Nature Genetics
New York, NY 10004, USA
www.nature.com/ng

Reviewers' Comments:

Reviewer #1:

Remarks to the Author:

The paper is much improved relative to the previous version and the significance of the developments reported here are now much easier to interpret.

This is an impressive technical achievement as the maize genome is similar in complexity to the human genome, and the authors have achieved a highly contiguous T2T assembly. Notably, this includes the rDNA arrays, which have been challenging to assemble in all species to date. The authors report a read-extension method to assemble the rDNA which is likely to be of general interest in the field of genomics.

There are also a number of remarkable findings concerning the repeat landscape – although in many cases deeper thought could be given to the biological significance.

The assembly of the rDNA arrays is a real achievement, and interestingly Copia were found within the 5S array and Gypsy in the 45S. As far as know, this arrangement is highly novel and has not been observed before (very few rDNA arrays have been assembled) - this is certainly worth commenting on and could receive more comment in the Discussion (also the abstract). Copia and gypsy are very generic names – could the authors please give more detail on which families are involved in each case. For example, I assume the Gypsy within the 45S do not include CRM? If not, which families are in there, and do they look like specific types? Same question applies to the Copia in the 5S - are they a specific type, or is it a similar spectrum to found elsewhere in the genome?

I mentioned this in my original review – but I do strongly feel the authors should include DNA

methylation analysis, and I feel their reasons for not doing so are unclear. The existing ONT data could easily be used for methylation calling. Excellent pipelines to do this exist – notably DeepSignal-plant, which provide excellent calls in CG, CHG and CHH contexts in plant data. Also, the paper focuses on the repeat component of the genome. As repeats are highly DNA methylated in plants, adding this data will greatly increase the impact and significance of the work. I would think this is well within the capabilities of the authors and not a big thing to ask that would be a great improvement on the MS. For example, in comparison the recent Arabidopsis long read assembly papers included methylation profiles that were informative. For example, CHG methylation was depleted around the CENH3 enriched regions, which would be interesting to compare with in maize.

Methylation profiles in intact rDNA arrays would also be highly novel and impactful.

On line 519, I strongly disagree with the authors that their assembly implies 'structurally highly stable' rDNA arrays. There is abundant evidence of duplication and transposition in the structures they report. It also seems hard to infer much about stability from a single genome. In fact, I expect that once other maize lines are assembled that the arrays will be structurally different – although this is purely speculation at this point. I strongly advise modifying this conclusion and instead noting that the complex patterns of rDNA copy duplications and transposon insertions points towards a complex history of repeat dynamics.

Thank you for adding the CENH3 ChIP-seq data which is a strong addition. Line 527 – please state how centromeres are being defined here. For the CentC absent centromeres – are the authors positive no other / alternative tandem repeats are there instead? As CRM are part of the Gypsy superfamily – could the authors explain the distinction between CRM and Gypsy that is often made eg lines 571 to 578 and in Fig 5C where they occupy different tracks. Do the Gypsy mentioned here lack the integrase chromo domain that typifies CRM for example?

Another reason to include DNA Methylation if for example on CEN8. It appears that CENH3 has recently migrated away from the repeat region located to the right. This is expected due to phenomena such as neocentromeres, and previously work in maize that has shown CENH3 'creeping' along chromosomes. It would be extremely interesting to know what is the methylation states of the now unoccupied repeats, and the new CENH3 location, particular the CHG context for the reasons discussed above.

Minor points:

Line 33 – change 'the very last structural 'dark matter' of the genome' to 'the repetitive component of the genome'. Its unclear how much is structural (and what that term means). Also it is no longer 'dark matter' as you have illuminated it - 'ex-dark matter'?

Line 36 – delete 'could'.

Line 61 – delete 'In fact' and merge sentences.

Overall the abstract feels fragmented – the sentences are not well connected, and the contribution/significance of the paper is hard to appreciate as it stands.

Line 348 – change 'uncomplete' to 'incomplete'.

Line 625 – by what criteria is it ‘the most complex’?

Reviewer #2:

Remarks to the Author:

In their revised version of the manuscript titled “A complete telomere-to-telomere assembly of maize genome” the authors responded to some of my comments regarding the validation of the assembly.

First, as I noticed in my first review, using the V1 assembly to organize the ONT long-read contigs is weird. Indeed, the authors select 19 large contigs and reject 548 contigs that were mostly classified as redundant (based on alignment with an incomplete assembly). The authors generated a lot of high-quality data but did not sequence Hi-C libraries which could have avoided the use as a backbone of the incomplete V1 assembly.

The authors claimed that “In plants, only the watermelon genome, with a size of 369 Mb and only several dozens of 45S rDNAs, has been completely assembled thus far”. I think the Arabidopsis and banana genomes are also complete, because the remaining gaps are located in regions with multi-copy repeats, where the copy number is arbitrary as shown in Mo17 with the rDNA and TAG repeat arrays.

As suggested, the authors use coverage with HiFi reads, but I think they should also add this coverage analysis to Extended Figures 17 and 18.

Also, regarding my previous comment “The combination of the ONT and HiFi assemblies, in particular to fill the gaps, ask questions. In gaps 4, 7 and 10, a large part of the ONT contigs did not align to HiFi contigs, and the authors removed those regions in the final assembly, if I understood the process well (first 200Kb of contig6, first 100Kb of contig 13 and first 250Kb of contig 19 in Extended Data Figure 3). Again, the authors should use external data to validate the correct filling of these gaps.”, I think the authors should make it clear that the removed parts of contigs were in fact redundant fragments. It should be checked easily by aligning the end of the contigs (e.g. the end of contig 5 and the start of contig 6 in Gap 4, Extended Figure 3).

Finally, the legend color of Figure 6B is not correct: orange in the legend and yellow in the figure.

Reviewer #3:

Remarks to the Author:

My concerns have been addressed except one: I would like to see the PacBio HiFi read coverage across the whole genome displayed in a figure, not just summarized in the text. This is important because the PacBio HiFi reads in theory provide a more independent source of data to test the quality of the assembly than the ONT reads.

Reviewer #4:

Remarks to the Author:

All of the points I raised previously have been sufficiently addressed. Thanks!

KM

Author Rebuttal, first revision:

Reviewer #1:

Remarks to the Author:

The paper is much improved relative to the previous version and the significance of the developments reported here are now much easier to interpret.

This is an impressive technical achievement as the maize genome is similar in complexity to the human genome, and the authors have achieved a highly contiguous T2T assembly. Notably, this includes the rDNA arrays, which have been challenging to assemble in all species to date. The authors report a read-extension method to assemble the rDNA which is likely to be of general interest in the field of genomics.

Response: Thank you for your encouraging comments.

There are also a number of remarkable findings concerning the repeat landscape – although in many cases deeper thought could be given to the biological significance.

The assembly of the rDNA arrays is a real achievement, and interestingly Copia were found within the 5S array and Gypsy in the 45S. As far as know, this arrangement is highly novel and has not been observed before (very few rDNA arrays have been assembled) - this is certainly worth commenting on and could receive more comment in the Discussion (also the abstract). Copia and gypsy are very generic names – could the authors please give more detail on which families are involved in each case. For example, I assume the Gypsy within the 45S do not include CRM? If not, which families are in there, and do they look like specific types? Same question applies to the Copia in the 5S - are they a specific type, or is it a similar spectrum to found elsewhere in the genome?

Response: Totally, there were 116.9 kb TEs (81.3% were Copia elements) in the 5S rDNA array and 343.8 kb TEs (83.8% were Gypsy elements) in the 45S rDNA array. The TEs in 5S and 45S rDNA arrays

are not a specific family of Copia or Gypsy elements. Compared to the flanking regions, TEs inserted in the 5S rDNA array were enriched with Opie (29.9%) and ji (16.0%) families of Copia elements. Remaining TEs in 5S rDNA array were mainly other types of Copia elements (35.5%), Gypsy_huck (11.3%), Gypsy_prem1 (6.8%). TEs in the 45S rDNA array were enriched with Prem1 (36.3%), Flip (21.5%), and Gyma (6.5%) families of Gypsy elements. Other types of Gypsy elements taken up 19.5% of TEs in 45S rDNA array. Yes, there was no CRM in the 45S array. These were added in the revision (lines 511 to 516, Extended Data Figure 26). The enrichment of Copia in 5S array and Gypsy in 45S array was commented in the Discussion (lines 652 to 654).

I mentioned this in my original review – but I do strongly feel the authors should include DNA methylation analysis, and I feel their reasons for not doing so are unclear. The existing ONT data could easily be used for methylation calling. Excellent pipelines to do this exist – notably DeepSignal-plant, which provide excellent calls in CG, CHG and CHH contexts in plant data. Also, the paper focuses on the repeat component of the genome. As repeats are highly DNA methylated in plants, adding this data will greatly increase the impact and significance of the work. I would think this is well within the capabilities of the authors and not a big thing to ask that would be a great improvement on the MS. For example, in comparison the recent Arabidopsis long read assembly papers included methylation profiles that were informative. For example, CHG methylation was depleted around the CENH3 enriched regions, which would be interesting to compare with in maize.

Methylation profiles in intact rDNA arrays would also be highly novel and impactful.

Response: Thank you for your suggestion. We can't agree more that it is significant to analysis DNA methylation patterns for repeat components of the genome, particularly for the centromeres and rDNA arrays. Following your suggestion, we have taken several months by trying to dig out some publishable results to be included in our revision. Unfortunately, due to several technical and time constraint, we have to ask for your favor that we will rather not report the methylation analyses in this manuscript.

Initially trained using human data, the methylation pipeline based on ONT data seems to be quite ready and straightforward. But in our experience, it could have a number of species-specific parameters needs to

be fine-tuned. In addition, precisely alignment of long-read ONT data in the highly repetitive regions is still a big challenge.

Knowing that we may not be able to include the methylation data in revision, we had particularly contacted the editor to explain the challenge we face. The editor has expressed the possibility of being flexible for the specific question. Here I am asking you a favor for being similarly flexible for this particular question.

Lastly, a personal timing consideration, as indicated in the Acknowledgement section, we want to dedicate this paper to my postdoc advisor, Joachim Messing (1947-2019), a pioneer in sequencing technology advancement and genome sequencing of maize and several other plant species. I personally hope that this dedicated paper can be published or at least officially accepted soon, so that it can be presented at the memorial symposium scheduled in earlier April.

On line 519, I strongly disagree with the authors that their assembly implies ‘structurally highly stable’ rDNA arrays. There is abundant evidence of duplication and transposition in the structures they report. It also seems hard to infer much about stability from a single genome. In fact, I expect that once other maize lines are assembled that the arrays will be structurally different – although this is purely speculation at this point. I strongly advise modifying this conclusion and instead noting that the complex patterns of rDNA copy duplications and transposon insertions points towards a complex history of repeat dynamics.

Response: Thank you for this very thoughtful suggestion. We revised this part according to your suggestion, pointing out the complex nature of rDNA duplications and TE insertions of the NOR, while not mentioning its less likely feature of stability.

Thank you for adding the CENH3 ChIP-seq data which is a strong addition. Line 527 – please state how centromeres are being defined here. For the CentC absent centromeres – are the authors positive no other / alternative tandem repeats are there instead? As CRM are part of the Gypsy superfamily – could the

authors explain the distinction between CRM and Gypsy that is often made eg lines 571 to 578 and in Fig 5C where they occupy different tracks. Do the Gypsy mentioned here lack the integrase chromo domain that typifies CRM for example?

Response: Here we determined the centromeric regions of the T2T Mo17 genome according to the enrichment level of CENH3, which was obtained via ChIP-seq with anti-CENH3 antibody (lines 524 to 526, see detail method in lines 1218 to 1234).

No CentC, no other satellites were identified in the CentC absent centromeres. We specifically note that no other satellite as well in the revision (line 547).

The question regarding whether the non-CRM Gypsy identified around centromere lack “the integrase chromo domain” (or the CR motif) that typifies CRM is highly insightful, the CR motif had been proposed to potentially have function in targeting CRM to the centromeric regions. To answer this, we made additional analyses. We found that most of the non-CRM Gypsy elements in centromeric regions do not have CR motif that typifies CRM. Among 166 intact non-CRM Gypsy elements, only 50 of them harbored with type II chromodomain (a general type of chromodomain for plant species, but distinct with CR motif), 108 of which having no chromodomain at all, neither the type II chromodomain nor the CR motif.

Another reason to include DNA Methylation if for example on CEN8. It appears that CENH3 has recently migrated away from the repeat region located to the right. This is expected due to phenomena such as neocentromeres, and previously work in maize that has shown CENH3 'creeping' along chromosomes. It would be extremely interesting to know what is the methylation states of the now unoccupied repeats, and the new CENH3 location, particular the CHG context for the reasons discussed above.

Response: Thank you for your great suggestion. In addition to our response for DNA methylation in earlier section, additional ChIP-seq data of histone modifications (such as K9me2 etc.) are needed to be analyzed along with the CHG methylation results. Indeed, it is very interesting to see the DNA

methylation and some histone modification level at both the CENH3 occupied neocentromeric region and that of the original CentC-rich region. I very sorry that we can not include this at this revision.

Minor points:

Line 33 – change ‘the very last structural ‘dark matter’ of the genome’ to ‘the repetitive component of the genome’. Its unclear how much is structural (and what that term means). Also it is no longer ‘dark matter’ as you have illuminated it - 'ex-dark matter'?

Response: Thank you. It’s done.

Line 36 – delete ‘could’.

Response: It’s done.

Line 61 – delete ‘In fact’ and merge sentences.

Response: It’s done.

Overall the abstract feels fragmented – the sentences are not well connected, and the contribution/significance of the paper is hard to appreciate as it stands.

Response: The abstract is now revised.

Line 348 – change ‘uncomplete’ to ‘incomplete’.

Response: It’s done.

Line 625 – by what criteria is it ‘the most complex’?

Response: The “most” is removed.

Reviewer #2:

Remarks to the Author:

In their revised version of the manuscript titled “A complete telomere-to-telomere assembly of maize genome” the authors responded to some of my comments regarding the validation of the assembly.

First, as I noticed in my first review, using the V1 assembly to organize the ONT long-read contigs is weird. Indeed, the authors select 19 large contigs and reject 548 contigs that were mostly classified as redundant (based on alignment with an incomplete assembly). The authors generated a lot of high-quality data but did not sequence Hi-C libraries which could have avoided the use as a backbone of the incomplete V1 assembly.

Response: Thank you for pointing out this. Sorry that we mistakenly referred the V1 for order the very last 19 contigs. In fact, only 19 contigs for 10 chromosomes, with less than three contigs for each chromosome, the anchoring or ordering is rather easy. Actually, their orders are rather obvious. A few genetic markers locating in each of the contigs can be sufficient to sort this out. The published high-density genetic map of maize containing approximately 4.4 million genetically mapped unique sequenced tags, much more than we needed, was used to construct the pseudomolecules. We revised this accordingly in our revision (lines 116 to 119).

The authors claimed that “In plants, only the watermelon genome, with a size of 369 Mb and only several dozens of 45S rDNAs, has been completely assembled thus far”. I think the Arabidopsis and banana genomes are also complete, because the remaining gaps are located in regions with multi-copy repeats, where the copy number is arbitrary as shown in Mo17 with the rDNA and TAG repeat arrays.

Response: We totally agree with your view that the genomes can be considered as complete in case that the remaining gaps are located mostly in regions with arbitrary number of specific repeats. We therefore rephrased this part of introduction. The paper reporting the high quality essentially completed banana genome has also been cited, together with that of the updated rice, Arabidopsis and Watermelon genomes.

As suggested, the authors use coverage with HiFi reads, but I think they should also add this coverage

analysis to Extended Figures 17 and 18.

Response: Coverage analysis based on PacBio HiFi reads was added to Extended Figures 17 and 18.

Also, regarding my previous comment “The combination of the ONT and HiFi assemblies, in particular to fill the gaps, ask questions. In gaps 4, 7 and 10, a large part of the ONT contigs did not align to HiFi contigs, and the authors removed those regions in the final assembly, if I understood the process well (first 200Kb of contig6, first 100Kb of contig 13 and first 250Kb of contig 19 in Extended Data Figure 3). Again, the authors should use external data to validate the correct filling of these gaps.”, I think the authors should make it clear that the removed parts of contigs were in fact redundant fragments. It should be checked easily by aligning the end of the contigs (e.g. the end of contig 5 and the start of contig 6 in Gap 4, Extended Figure 3).

Response: Thank you for your suggestion. We checked the ONT reads coverage for the end of the contigs of gaps 4, 7, 10. Distinct to the normal gap ends (which were remained in the final assembly) with gradually decreased coverage, almost no aligned ONT reads were observed for the regions removed in the final assembly, which suggested that those removed parts of contigs were in fact redundant or misassembled fragments, and further confirmed the assemblies of these gap-closed regions. Please see Figure 1C and Extended Figure 6.

Finally, the legend color of Figure 6B is not correct: orange in the legend and yellow in the figure.

Response: The legend was corrected.

Reviewer #3:

Remarks to the Author:

My concerns have been addressed except one: I would like to see the PacBio HiFi read coverage across the whole genome displayed in a figure, not just summarized in the text. This is important because the PacBio HiFi reads in theory provide a more independent source of data to test the quality of the assembly than the ONT reads.

Response: The PacBio HiFi read coverage across the whole genome has been added in the revision. Please see line 315 and Extended Figure 17.

Reviewer #4:

Remarks to the Author:

All of the points I raised previously have been sufficiently addressed. Thanks!

KM

Decision Letter, second revision:

17th Mar 2023

Dear Dr. Lai,

Thank you for submitting your revised manuscript "A complete telomere-to-telomere assembly of the maize genome" (NG-A59846R1). It has now been seen by the original referees and their comments are below. The reviewers find that the paper has improved in revision, and therefore we'll be happy in principle to publish it in Nature Genetics, pending minor revisions to satisfy the referees' final requests and to comply with our editorial and formatting guidelines.

Sincerely,

Wei Li, PhD
Senior Editor
Nature Genetics
New York, NY 10004, USA
www.nature.com/ng

Reviewer #1 (Remarks to the Author):

Thank you - I appreciate the ONT derived methylation data will take further work, which I look forward to reading about

Reviewer #2 (Remarks to the Author):

All the concerns I raised have been now addressed.

Reviewer #3 (Remarks to the Author):

Now that the new Extended Data Figure 17 shows coverage with both HiFi and ONT reads over the entire assembly, it does indeed support the claim of “uniform coverage across nearly all genomic regions”. However, the exceptional regions in the figure, with non-uniform coverage (CentC arrays, knob arrays, and potentially rDNA arrays), may reveal an unpublished artefact of the assembly method. These tandem repeats appear to have higher abundance in the genome assembly than in the source reads used to create the assembly. In other words, they have been over assembled. This interpretation would be consistent with recent evidence I have seen that ONT based assemblies over-assemble tandem repeats into larger arrays than actually exist (unpublished work). The problem might be more evident when mapping HiFi reads than ONT reads back to the genome reads because the ONT reads contributed more to the erroneous part of the assembly than HiFi reads did. It would be a service to the community if this dis-concordance between tandem repeat abundance between assembly and source reads were explicitly acknowledged. The abstract should be modified to acknowledge that assembled tandem repeat arrays may not actually allow one to “precisely dissect the repeat compositions” after all. While this may appear to be a disappointing result in that it suggests the assembly quality in these regions is still inaccurate, it increases the significance of the paper in terms of correcting what may be a widespread problem in interpreting recent genome assemblies.

Author Rebuttal, second revision:

Reviewer #1:

Remarks to the Author:

Thank you - I appreciate the ONT derived methylation data will take further work, which I look forward to reading about

Reviewer #2:

Remarks to the Author:

All the concerns I raised have been now addressed.

Reviewer #3:

Remarks to the Author:

Now that the new Extended Data Figure 17 shows coverage with both HiFi and ONT reads over the entire assembly, it does indeed support the claim of “uniform coverage across nearly all genomic regions”. However, the exceptional regions in the figure, with non-uniform coverage (CentC arrays, knob arrays, and potentially rDNA arrays), may reveal an unpublished artefact of the assembly method. These tandem repeats appear to have higher abundance in the genome assembly than in the source reads used to create the assembly. In other words, they have been over assembled. This interpretation would be consistent with recent evidence I have seen that ONT based assemblies over-assemble tandem repeats into larger arrays than actually exist (unpublished work). The problem might be more evident when mapping HiFi reads than ONT reads back to the genome reads because the ONT reads contributed more to the erroneous part of the assembly than HiFi reads did. It would be a service to the community if this dis-concordance between tandem repeat abundance between assembly and source reads were explicitly acknowledged. The abstract should be modified to acknowledge that assembled tandem repeat arrays may not actually allow one to “precisely dissect the repeat compositions” after all. While this may appear to be a disappointing result in that it suggests the assembly quality in these regions is still inaccurate, it increases the significance of the paper in terms of correcting what may be a widespread problem in interpreting recent genome assemblies.

Response: We agree this potential limitation the reviewer raised. Duo to the space limit of abstract, we add additional discussion at the discussion section to point out that it is technological difficulty at present to precisely dissection of the exact copy number of highly tandemly duplicated sequences at the extraordinary repetitive regions for particularly tissues or cell types.

Final Decision Letter:

5th May 2023

Dear Dr. Lai,

I am delighted to say that your manuscript "A complete telomere-to-telomere assembly of the maize genome" has been accepted for publication in an upcoming issue of Nature Genetics.

After the grant of rights is completed, you will receive a link to your electronic proof via email with a

request to make any corrections within 48 hours. If, when you receive your proof, you cannot meet this deadline, please inform us at rjsproduction@springernature.com immediately.

Your paper will be published online after we receive your corrections and will appear in print in the next available issue. You can find out your date of online publication by contacting the Nature Press Office (press@nature.com) after sending your e-proof corrections. Now is the time to inform your Public Relations or Press Office about your paper, as they might be interested in promoting its publication. This will allow them time to prepare an accurate and satisfactory press release. Include your manuscript tracking number (NG-A59846R2) and the name of the journal, which they will need when they contact our Press Office.

Please note that *Nature Genetics* is a Transformative Journal (TJ). Authors may publish their research with us through the traditional subscription access route or make their paper immediately open access through payment of an article-processing charge (APC). Authors will not be required to make a final decision about access to their article until it has been accepted. [Find out more about Transformative Journals](https://www.springernature.com/gp/open-research/transformative-journals)

Authors may need to take specific actions to achieve [compliance with funder and institutional open access mandates](https://www.springernature.com/gp/open-research/funding/policy-compliance-faqs). If your research is supported by a funder that requires immediate open access (e.g. according to [Plan S principles](https://www.springernature.com/gp/open-research/plan-s-compliance)) then you should select the gold OA route, and we will direct you to the compliant route where possible. For authors selecting the subscription publication route, the journal's standard licensing terms will need to be accepted, including [self-archiving and license to publish](https://www.nature.com/nature-portfolio/editorial-policies/self-archiving-and-license-to-publish). Those licensing terms will supersede any other terms that the author or any third party may assert apply to any version of the manuscript.

Please note that Nature Portfolio offers an immediate open access option only for papers that were

first submitted after 1 January, 2021.

If you have not already done so, we invite you to upload the step-by-step protocols used in this manuscript to the Protocols Exchange, part of our on-line web resource, natureprotocols.com. If you complete the upload by the time you receive your manuscript proofs, we can insert links in your article that lead directly to the protocol details. Your protocol will be made freely available upon publication of your paper. By participating in natureprotocols.com, you are enabling researchers to more readily reproduce or adapt the methodology you use. [Natureprotocols.com](http://natureprotocols.com) is fully searchable, providing your protocols and paper with increased utility and visibility. Please submit your protocol to <https://protocolexchange.researchsquare.com/>. After entering your [nature.com](http://www.nature.com) username and password you will need to enter your manuscript number (NG-A59846R2). Further information can be found at <https://www.nature.com/nature-portfolio/editorial-policies/reporting-standards#protocols>

Sincerely,
Wei

Wei Li, PhD
Senior Editor
Nature Genetics
New York, NY 10004, USA
www.nature.com/ng